# Improving permafrost physics in the coupled Canadian Land Surface Scheme (v. 3.6.2) and Canadian Terrestrial Ecosystem Model (v. 2.1) (CLASS-CTEM)

Joe R. Melton[1], Diana L. Verseghy[2,*], Reinel Sospedra-Alfonso[3], and Stephan Gruber[4]

[1]Climate Research Division, Environment and Climate Change Canada, Victoria, B.C., Canada
[2]Formerly at Climate Research Division, Environment and Climate Change Canada
[3]Canadian Centre for Climate Modelling and Analysis, Climate Research Division, Environment and Climate Change Canada, Victoria, B.C., Canada
[4]Department of Geography and Environmental Studies, Carleton University, Ottawa, Canada
[*]Retired

**Correspondence:** Joe R. Melton (joe.melton@canada.ca)

**Abstract.** The Canadian Land Surface Scheme and Canadian Terrestrial Ecosystem Model (CLASS-CTEM) together form the land surface component of the Canadian Earth System model (CanESM). Here we investigate the impact of changes to CLASS-CTEM that are designed to improve the simulation of permafrost physics. Eighteen tests were performed including changing the model configuration (number and depth of ground layers, different soil permeable depth datasets, adding a surface moss layer), and investigating alternative parameterizations of soil hydrology, soil thermal conductivity and snow properties. To evaluate these changes, CLASS-CTEM outputs were compared to 1570 active layer thickness (ALT) measurements from 97 observation sites that are part of the Global Terrestrial Network for Permafrost (GTN-P), 105 106 monthly ground temperature observations from 132 GTN-P borehole sites, a blend of 5 observation-based snow water equivalent (SWE) datasets (Blended-5), remotely-sensed albedo, and seasonal discharge for major rivers draining permafrost regions. From the tests performed, the final revised model configuration has more ground layers (increased from 3 to 20) extending to greater depth (from 4.1m to 61.4 m) and uses a new soil permeable depths dataset with a surface layer of moss added. The most beneficial change to the model parameterizations was incorporation of unfrozen water in frozen soils. These changes to CLASS-CTEM cause a small improvement in simulated SWE with little change in surface albedo but greatly improve the model performance at the GTN-P ALT and borehole sites. Compared to the GTN-P observations, the revised CLASS-CTEM ALTs have a weighted mean absolute error (wMAE) of 0.41 - 0.47 m (depending on configuration), improved from > 2.5 m for the original model, while the borehole sites see a consistent improvement in wMAE for most seasons and depths considered, with seasonal wMAE values for the shallow surface layers of the revised model simulation at most 3.7 °C, which is 1.2 °C more than the wMAE of the screen-level air temperature used to drive the model as compared to site-level observations (2.5 °C). Sub-grid heterogeneity estimates were derived from the standard deviation of ALT on the 1 km$^2$ measurement grids at the GTN-P ALT sites, the spread in wMAE in grid cells with multiple GTN-P ALT sites, as well as from 35 boreholes measured within a 1200 km$^2$ region as part of the Slave Province Surficial Materials and Permafrost Study. Given the size of the model grid cells (ca. 2.8°), sub-grid heterogeneity makes it likely difficult to appreciably reduce the wMAE of ALT or borehole temperatures much further.

## 1  Introduction

Permafrost underlies between 9 and 14 % of the exposed land surface north of 60°S (13 - 18 $\times 10^6$ km$^2$; Gruber, 2012). The presence of perennially frozen soil at depth has strong impacts on local hydrology, energy fluxes, plant communities, and carbon dynamics. Several factors influence ground temperature and therefore the presence of permafrost, including snow cover, vegetation structure and function, hydrology, and topography (Loranty et al., 2018). Permafrost has been warming and active layers have thickened over the last three decades (Vaughan et al., 2013). This trend is expected to continue due to climate change (Chadburn et al., 2017) making the carbon presently contained in frozen soils vulnerable to release to the atmosphere either as carbon dioxide or methane, depending on local conditions. Since the carbon stored in frozen soils becomes readily accessible to microbial respiration once soils thaw, accurately simulating the physics of the permafrost response to a changing climate is vital for reliable predictions of the permafrost carbon feedback to climate change.

The Canadian Land Surface Scheme (CLASS) is the land surface component of the Canadian Earth System Model (CanESM). CLASS has been tested for its cold regions performance in several studies previously. Tilley et al. (1997) evaluated CLASS (v. 2.5) at a site on the North Slope of Alaska. The principal conclusions of the study were that CLASS was most sensitive to ground column depth and soil composition with lesser sensitivity to variations in the radiative fluxes, specification of the overlying vegetation and the initial soil moisture. Bellisario et al. (2000) tested CLASS at a fen wetland and a willow-birch forest in the northern Hudson Bay lowlands. They found the upper soil layer temperatures to be consistently overestimated using the model's default mineral soil parameterization, whereas using the organic soil parametrization of Letts et al. (2000) improved the simulated temperatures significantly. Lafleur et al. (2000) did some tests with a subarctic open woodland site in Churchill, Manitoba using CLASS with the Letts et al. (2000) parameterization. Recommendations from their work included introducing a non-vascular plant functional type (PFT) and a sparse canopy representation, varying the minimum stomatal conductance according to PFT, and re-examination of the snow melt algorithm. The snow melt recommendations were subsequently investigated by Bartlett et al. (2006) and Brown et al. (2006). More recently, Paquin and Sushama (2014) used CLASS (v. 3.5) in the Canadian Regional Climate Model version 5 (CRCM5) to look at the impact of snow and soil parameterizations on simulated permafrost and climate. Their simulations included offline tests using the ERA-Interim meteorological forcing over the pan-Arctic region. Paquin and Sushama tested several options that have previously been made available in CLASS, but not yet implemented operationally, including, 1) increasing the number and depth of soil layers (47 levels extending to 65 m), 2) using the Letts et al. (2000) parameterization for peatlands and assuming an organic surface soil layer for most other regions, and 3) changing the snow thermal conductivity parameterization from Mellor (1977) to Sturm et al. (1997). The Sturm et al. formulation was subsequently adopted in CLASS v. 3.6 (Verseghy, 2017). Ganji et al. (2015) also used CLASS in

CRCM5 to investigate cold region hydrological performance. They reported improvements by incorporating supercooled soil water, fractional permeable area, and a changed hydraulic conductivity formulation for frozen soil. MacDonald (2015) coupled CLASS v. 3.6 to the Prairie Blowing Snow Model (PBSM) to simulate the influence of chinooks (Föhn winds) over the South Saskatchewan River Basin. He investigated 15 alternative parameterizations relating to the model physics and concluded by

recommending that four of those be considered for adoption in CLASS to improve the simulated snow water equivalent (SWE) and soil water. Three of the suggested parameterizations dealt with snow properties and the fourth related to soil thermal conductivity (MacDonald, 2015).

Our study evaluates the individual and combined effects of suggested enhancements to the Canadian Land Surface Scheme coupled to the Canadian Terrestrial Ecosystem Model (CLASS-CTEM) for simulating processes relevant to soils with per-

mafrost or pronounced seasonal freezing. The model enhancements suggested above have previously been recommended in research studies but not been previously implemented into the CLASS-CTEM framework (unless otherwise noted). Here we investigate the impact of these previously proposed model enhancements as well as several model configuration changes suggested in the literature. Based on this evaluation, a revised version of CLASS-CTEM, containing several enhancements is described and also evaluated. To evaluate model behaviour we draw upon measurements of the thickness of annual thaw in

perennially frozen soils (active layer thickness) and borehole temperature sites from the Global Terrestrial Network for Permafrost (GTN-P, 2016) along with other observation-based datasets for snow, surface albedo and runoff.

Numerous studies have investigated the permafrost physics performance of models (e.g. see review in Riseborough et al., 2008) including other large scale models used in ESM applications, such as JULES (Chadburn et al., 2015a, b), JS-BACH (Ekici et al., 2014), and the Community Land Model (CLM, e.g., Alexeev et al., 2007; Lawrence et al., 2008; Lee et al., 2014)

allowing us to design our proposed experiments based on their conclusions. The performance of CLASS-CTEM permafrost physics will be evaluated through offline simulations where the model is forced with reanalysis meteorology to avoid biases found in the simulated climate of the coupled model as well as biases in the associated feedbacks. This study is focused on model performance at the large spatial scale of the CanESM as our principle aim is to improve the simulated permafrost physics so that the carbon cycle processes in these regions is well bounded. It is therefore not aimed at shedding light on physical

processes in permafrost zones or investigating model performance at individual point locations as the model performance at a single site does not directly translate to model performance over large regions.

In the remainder of the paper, Section 2 describes the CLASS-CTEM model, the study design as well as parameterizations tested, and the GTN-P sites used in model evaluation. Section 3 evaluates the model performance as well as discussing the influence of sub-grid heterogeneity while Section 4 gives overall conclusions and discusses limitations of our study and future

directions for CLASS-CTEM development.

## 2 Experimental setup

### 2.1 CLASS-CTEM

CLASS (v. 3.6.2; Verseghy, 2017) coupled with CTEM (v.2.1; Melton and Arora, 2016) forms the land surface component of the CanESM. CLASS performs the land surface energy and water balance calculations on a, typically, half-hour timestep.

The model uses leaf area index (LAI), rooting depth, canopy mass, and vegetation height to evaluate the energy and water balance terms of the vegetation canopy and its interactions with the atmosphere. The number of soil layers can vary depending on the application but the standard model setup uses three soil layers of 0.1, 0.25, and 3.75 m thickness. The soil texture (sand, clay, organic matter) dataset used by CLASS-CTEM is the Global Soil Dataset for use in Earth System Models (GSDE; Shangguan et al., 2014). The soil permeable depth is from Zobler (1986) (hereafter Zobler86).CLASS v.3.6.2 adopts the soil

albedo approach of Lawrence and Chase (2007) with the incorporation of a soil colour index geophysical field.

CLASS prognostically determines the water content (liquid and frozen) and temperature of all soil layers at each timestep. Also calculated at each timestep, depending on ambient conditions, are the temperature, mass, albedo, and density of a single layer snow pack, interception of rain and snow on the vegetation canopy, and amount of ponded water on the soil surface. Mineral soils are parameterized using the pedotransfer functions of Cosby et al. (1984) and Clapp and Hornberger (1978).

Organic soils (organic matter >30% by weight) are modelled as peat following Letts et al. (2000). In the standard CLASS-CTEM framework, lateral transfers of heat or moisture between grid cells are neglected; the treatment of processes such as streamflow and blowing snow require the inclusion of separate, specialized routines (e.g., Soulis et al., 2000; Arora et al., 2001; MacDonald, 2015). All simulations presented here have no geothermal heat flux at the bottom of the soil column.

CTEM calculates the carbon and vegetation dynamics on a daily timestep receiving from CLASS daily mean soil moisture,

soil temperature, and net radiation. Photosynthesis and canopy conductance occur on the CLASS timestep. CTEM simulates the respiratory costs and carbon uptake for nine plant functional types (PFTs) which are subsets of the four CLASS PFTs. The CLASS PFTs (with corresponding CTEM PFTs in parentheses) are needleleaf tree (needleleaf deciduous and needleleaf evergreen), broadleaf tree (broadleaf cold deciduous, broadleaf drought/dry deciduous, and broadleaf evergreen), crop (photosynthetic pathway $C_3$ and $C_4$), and grass ($C_3$ and $C_4$). CTEM carries five carbon pools representing plant leaves, roots, and

stems along with two detrital pools for litter and soil C.

For global simulations, CLASS-CTEM is typically run at the CanESM atmosphere resolution which is approximately 2.8° by 2.8° corresponding to a grid cell size of approximately 49 000 km$^2$ at 45° latitude and about 33 500 km$^2$ at 70°. Various studies have used observation-based datasets to evaluate CLASS-CTEM at scales from site-level to global (e.g., Peng et al., 2014; Melton and Arora, 2014, 2016). While CLASS-CTEM is capable of running in a mosaic (multiple tiles per grid cell)

configuration (e.g. Melton and Arora, 2014; Melton et al., 2017), the simulations presented here are run with a single tile per grid cell.

## 2.2 Study design

Eighteen experiments were run to assess the impact of model geophysical fields (soil texture, soil permeable depth, and meteorological forcing), model setup (number of soil layers, addition of a moss layer), and model parameterizations (Table 1). The physical quantities used for model evaluation are presented in the next section. The initial model version (Exp. *Base model*) uses 3 ground layers of thicknesses 0.1, 0.25, and 3.75 m for a total depth of 4.1 m. The first seven experiments address model configuration and input geophysical fields. To test the sensitivity of simulated permafrost to meteorological forcing, CLASS-CTEM was forced with two different meteorological datasets, the Climate Research Unit - National Centres for Environmental Prediction (CRUNCEP v. 8; Viovy, 2016) and the Climate Research Unit - Japanese 55 year Reanalysis (CRUJRA55 v. 1.0.5; Harris et al., 2014; Kobayashi et al., 2015). CRUNCEP was used as the base forcing dataset with additional runs performed for some experiments with CRUJRA55 (see Table 1). While both of these meteorological datasets use the CRU TS dataset (Harris et al., 2014) as the underlying monthly climatology, they differ in their meterological models (NCEP or JRA55). Additionally the spatial resolution of JRA55 is $0.5°$ while that of NCEP is $2.5°$. Thus, the two datasets differ in their spatial and high frequency (sub-monthly) temporal variability. However these differences will be somewhat lessened by their regridding to the CLASS-CTEM model resolution. The meteorological inputs (surface air temperature, surface pressure, specific humidity, wind speed, precipitation, and longwave and shortwave radiation) are disaggregated from 6 hourly to half-hourly time steps while the simulation runs following the methodology in Melton and Arora (2016). Both datasets are available over the extended periods necessary for permafrost simulation (CRUNCEP v. 8: 1901 - 2016, CRUJRA55 v. 1.0.5: 1901 - 2017).

Exp. *20 ground layers* changes the number of ground layers from 3 to 20. The 20 layers have higher resolution near the surface with thicker layers at depth (see Table A1). If the permeable soil depth is shallower than the modelled ground column, layers below the soil permeable depth are treated like hydrologically inactive bedrock and are assigned thermal conductivity ($2.5$ W m$^{-1}$ K$^{-1}$) and heat capacity ($2.13 \times 10^6$ J m$^{-3}$ K$^{-1}$) values characteristic of sand particles (Verseghy, 2017). If the transition from permeable soil to impermeable bedrock occurs within a soil layer, CLASS calculates the water fluxes only in the depth of permeable soil but simulates one soil temperature for the layer.

The influence of the soil permeable depth dataset is examined by replacing the soil permeable depths of Zobler86 with either the SoilGrids dataset (Exp. *SoilGrids depth*,  Shangguan et al., 2017) or that of Pelletier et al. (2016) (hereafter referred to as Pel16; Exp. *Pel16 depth*). The influence of a moss layer is examined in Exp. *SoilGrids+Moss* and *Pel16+Moss*. In these experiments the top soil layer is replaced with photosynthetically-inactive moss with a higher porosity, hydraulic conductivity and heat capacity than mineral soil following Wu et al. (2016) (described in Appendix A1).

Whereas the first series of experiments just described investigated aspects of the model setup, the second series of experiments investigates alternative parameterizations and uses the *SoilGrids+Moss* experiment as a starting point (the same geophysical fields and model configuration). The alternative parameterizations are described in detail in Appendix sections A2 to A7. Briefly, these experiments fall into three main areas related to: 1) heat transfer, 2) snow, and 3) hydrology. The heat transfer experiments replace CLASS-CTEM's default soil thermal conductivity parameterization (Côté and Konrad, 2005) with that of de Vries (1963) following the recommendations of MacDonald (2015)(Exp. *deVries thermal cond.* results are

**Table 1.** List of experiments and the associated model theme they relate to. Experiments denoted with an asterisk were run with both the CRUNCEP and the CRUJRA55 meteorological forcing datasets.

| Experiment name | Theme | Description | Starting model configuration |
|---|---|---|---|
| Base model | | Original model setup | |
| 20 ground layers | Configuration | Twenty ground layers to a maximum depth of 61.5 m (see Table A1) | Base model |
| SoilGrids depth | Configuration | Soil permeable depth geophysical field (SoilGrids; Shangguan et al., 2017) | 20 ground layers |
| Pel16 depth | Configuration | Soil permeable depth geophysical field (Pel16; Pelletier et al., 2016) | 20 ground layers |
| SoilGrids+Moss* | Configuration | SoilGrids depth setup with first soil layer treated as non-photosynthetic moss layer following Wu et al. (2016) | SoilGrids depth |
| Pel16+Moss* | Configuration | Setup as above but with Pel16 depths | Pel16 depth |
| deVries thermal cond.* | Heat transfer | Soil thermal conductivity following de Vries (1963) | SoilGrids+Moss |
| Tian16 thermal cond. | Heat transfer | Soil thermal conductivity following Tian et al. (2016) | SoilGrids+Moss |
| Snow cover:Yang97* | Snow | Snow depth to fractional snow cover relation following Yang et al. (1997) | SoilGrids+Moss |
| Snow cover:Brown03 | Snow | Snow depth to fractional snow cover relation following Brown et al. (2003) | SoilGrids+Moss |
| Fresh snow density | Snow | Fresh snow density based on air temperature and wind speed following CROCUS as detailed in Essery et al. (1999) with a minimum density of 50 kg m$^{-3}$ following MacDonald (2015) | SoilGrids+Moss |
| Snow albedo decay | Snow | Efficient spectral snow albedo decay (Dickinson et al., 1993) | SoilGrids+Moss |
| Super-cooled water | Hydrology | Unfrozen water in frozen soils (super-cooled soil water) following Niu and Yang (2006) | SoilGrids+Moss |
| Modif. hydrology | Hydrology | Soil matric potential and effective saturated conductivity are modified for the influence of frozen water as described in Ganji et al. (2015) | SoilGrids+Moss |

discussed in the Supplement). As de Vries (1963) does not account for frozen water in soil, whereas Côté and Konrad (2005) does, a further experiment uses a recently published parameterization that simplifies and extends de Vries (1963) to include both frozen and unfrozen water (Exp. *Tian16 thermal cond.*; See Section A2; Tian et al., 2016). Four experiments were devoted to aspects of how snow is simulated in CLASS-CTEM. Experiments *Snow cover:Yang97* and *Snow cover: Brown03* replace CLASS-CTEM's default function to relate snow depth to grid cell fractional snow cover from a linear relationship (Verseghy, 2017) to a hyperbolic tangent (following Yang et al., 1997) or an exponential function (following Brown et al., 2003), respectively (Supplement Figure 2). Another experiment (*Fresh snow density*) changed the calculation for the density of freshly-fallen snow from one based solely on air temperature (Verseghy, 2017) to also considering wind speed following the CROCUS model (Essery et al., 1999). The final experiment concerned with aspects of the snow parameterization is *Snow*

*albedo decay*. CLASS-CTEM uses an empirical exponential decay function to simulate the decrease in snow albedo as snow ages. In *Snow albedo decay*, the default parameterization is replaced by an efficient spectral method (Wiscombe and Warren, 1980; Dickinson, 1983). The last series of experiments looked at hydrology. Water in soils can be, partially or completely, unfrozen at temperatures below 0°C due to the effects of interfacial curvature, adsorption forces and solutes (Watanabe and Mizoguchi, 2002; Dall'Amico et al., 2011). Experiment *Super-cooled water* incorporated the unfrozen water in frozen soils parameterizaton of Niu and Yang (2006) and Exp. *Modif. hydrology* modifies the soil matric potential and saturated hydraulic conductivity to account for the influence of frozen water following Ganji et al. (2015).

For model spinup, the meteorological forcing years of 1901 – 1925 were cycled over repeatedly until the model reached active layer thickness (ALT) equilibrium (less than 0.05 m difference between average ALT between spinup cycles across all cells with permafrost within them). To run from 1851 to 2016 while atmospheric CO2 concentration and land cover evolved, the climate was cycled over twice from 1901 to 1925 for the years 1851–1900, then the model climate was allowed to run freely from 1901 to 2016. For the simulations presented here, CLASS-CTEM was run with a prescribed, rather than prognostically determined distribution of PFTs.

Active layer thickness in CLASS-CTEM is determined by the temperature and water content of the ground layers. If a layer's temperature is 0°C, the frozen water fraction is used to estimate the thickness of freezing within the layer, i.e., if half of the water content in the layer is frozen, the ALT is assumed to be halfway through the layer. Permafrost area in the model domain was calculated by selecting grid cells with active layer thicknesses less than the model total ground column and multiplying by the grid cell area.

## 2.3 Datasets used for model evaluation

### 2.3.1 Active layer thickness sites from the Global Terrestrial Network for Permafrost (GTN-P)

To evaluate CLASS-CTEM, 97 open access GTN-P ALT sites were chosen due to their locations in regions of continuous or discontinuous permafrost (GTN-P, 2016; Biskaborn et al., 2019) (accessed May 11[th], 2017; Supplement Table 1 and Figure 1). No sites in areas of sporadic or isolated permafrost were used due to the difficulty in representing this type of permafrost within a large model grid. While we attempted to have as broad a spatial coverage of the GTN-P sites as possible, no open access sites were available for Eastern Canada and Fennoscandia. For comparison with CLASS-CTEM, at each observation time, the average of the sampling grid was determined at each GTN-P ALT site. Then for each site, the sampling grid averages were converted to monthly mean values. The closest grid cell was determined from the centre of the model grid cells to the ALT sampling location and the modelled monthly average ALTs were compared to the observed values. This resulted in the 97 GTN-P sites, with 1570 ALT observations, being placed into 37 CLASS-CTEM grid cells. As multiple GTN-P sites can be co-located in one CLASS-CTEM grid cell, the weighted mean absolute error (wMAE) for a grid cell was found by averaging the MAE calculated at each site situated within one CLASS-CTEM grid cell.

### 2.3.2 Borehole temperatures from the GTN-P

Borehole data from the GTN-P were downloaded for 132 open-access sites found in the permafrost (including continuous, discontinuous, sporadic, and isolated) or permafrost-free domains (accessed May $11^{th}$, 2017). Most of the boreholes are in Eurasia with few in North America (Figure 1; Supplement Table 2). Each site has its own unique time period of observations and number and depth of observations. At each site, the depths of borehole temperatures were selected to be within 0.05 and 3.0 m of the ground surface and the observations were averaged to monthly values. For each borehole and each observation depth, the CLASS-CTEM output was selected for the nearest grid cell and the same month as the observations. Linear interpolation was then used to determine the simulated soil temperature for the same soil depth as the observation. As with the ALT sites, several steps were needed to avoid biasing the comparison with CLASS-CTEM. First, borehole sites co-located in the same CLASS-CTEM grid cell were flagged. The 132 borehole sites are located in 73 unique CLASS-CTEM grid cells. Secondly, the number of observations varied by borehole site so when calculating the kernel density estimates (KDE; presented later) within a model grid cell each observation was weighted by the total number of observations per grid cell. Thus grid cells with many GTN-P borehole sites will have each observation weighted less than sites with fewer observations so each grid cell contributes equally to the KDE estimation and the calculation of wMAE.

### 2.3.3 Snow, albedo and runoff

Snow water equivalent (SWE) from CLASS-CTEM is compared to the Blended-5 dataset for the period from January 1981 to December 2010. Blended-5 is a multi-dataset SWE product developed by Mudryk et al. (2015) that combines five observation-based SWE datasets. Our analysis is limited to regions northward of 45°N with climatological SWE > 4 mm to avoid regions of ephemeral snow. Simulated land surface albedo is compared to the MODIS MCD43C3 white-sky albedo (MODIS Adaptive Processing System, NASA, 2016) for the period spanning February 2000 to December 2013. Similar to SWE, we limit our analysis to regions northward of 45°N. We compared our simulated seasonal runoff to measured discharge rates for seven major river basins that drain permafrost regions for the period from 1965 to 1984 (Ob, Volga, Lena, Yenisei, Yukon, Mackenzie, and Amur rivers; UNESCO Press, 1993). This comparison is limited to seasonal discharges since the CLASS-CTEM runoff is not routed, thus the timing of transport of the water from each grid cell to the river mouth is neglected. On a seasonal timescale this should not cause serious errors but the results must be interpreted with caution.

### 2.3.4 Permafrost distributions from the literature

Because permafrost cannot easily be observed spatially and reliable data are sparse, global or continental-scale simulation results are often compared to estimates of permafrost distributions. Most prominently, this is the "Circum-Arctic map of permafrost and ground-ice conditions" (Brown et al., 1997) that distinguishes zones of permafrost extent at a scale of 1:10 000 000. These zones are based on expert assessment and manual delineation, often following isotherms of mean annual air temperature. Here, we use *permafrost extent* to refer to the fraction (0 - 1) of the surface that is underlain by permafrost within a pixel or a polygon, *permafrost area* to the actual area (km$^2$) underlain by permafrost and *permafrost region* is used to denote the area

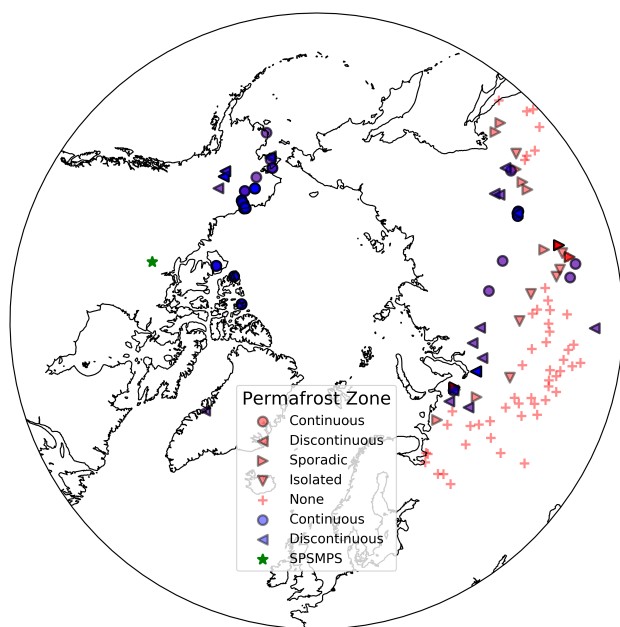

**Figure 1.** Locations of the 97 GTN-P ALT sites (blue; Supplement Table 1), 132 GTN-P borehole observation sites (red; Supplement Table 2) and the Slave Province Surficial Materials and Permafrost Study (green; SPSMPS; Lac de Gras, Northwest Territories, Canada) used for model evaluation. Each site is classified according to its permafrost zone listed in the GTN-P. The site markers are semi-transparent hence regions with many closely located GTN-P sites will cause overlap, and darkening, of the markers.

($km^2$) where some proportion of the ground can be expected to contain permafrost. The permafrost region is commonly taken to include areas with a permafrost extent exceeding some threshold (Zhang et al., 2000; Gruber, 2012). These definitions are relevant because CLASS-CTEM produces a binary result, i.e., permafrost is present or absent in a cell, and the classes (Zhang et al., 2000; Brown et al., 1997) and the continuous index (Gruber, 2012) of permafrost extent that are used for comparison need to be interpreted appropriately. Neglecting aggregation effects (Giorgi and Avissar, 1997), which arise when the average fine-scale behaviour of a simulated environmental variable is not equal to the simulated coarse-scale behaviour, a threshold of permafrost extent at 50% provides a first estimate of the region that should be compared with a model producing a binary result. For example, environmental conditions that give rise to a permafrost extent of 60% would likely be considered to have permafrost in the binary model and their area would be counted as having permafrost entirely (rather than only 60% of it). Similarly, conditions that produce a permafrost extent of 40% would likely result in not having permafrost in a binary model. As a consequence, we use the total area of all polygons or pixels with an expected permafrost extent larger than 50% as the appropriate area to compare with the results from CLASS-CTEM, termed 'region_50'. This includes continuous and extensive discontinuous permafrost in the Brown et al. map totalling to 15 Mkm$^2$ (Zhang et al. 1999) and a similar number can be interpreted from a plot of permafrost zonation index and permafrost region (Gruber, 2012).

## 3 Results and Discussion

### 3.1 Comparison against GTN-P ALT sites: sites with no simulated permafrost

A first simple test of permafrost performance for CLASS-CTEM is to check whether the GTN-P ALT sites are in fact simulated as containing permafrost. Given that CLASS-CTEM is being run on the CanESM grid (ca. 2.8°), it is possible that site conditions such as meteorology, orography, or vegetation at the GTN-P ALT measurement sites could be quite dissimilar to those of the nearest grid cell, which covers many thousands of $km^2$. In such cases, CLASS-CTEM could simulate no permafrost where some permafrost indeed exists. Per experiment, the number of sites with no permafrost simulated are listed in Table 2. These ALT sites were removed from further analysis as the ALT in sites without permafrost is not defined. Most experiments had between six and eight observation sites (corresponding to 4 to 6 grid cells) incorrectly simulated as permafrost-free (ISPF). The *Base model* experiment has significantly more sites ISPF at 15, corresponding to 2 or 3 additional grid cells. In general, for the same experiment, the CRUJRA55 meteorological forcing results in fewer grid cells ISPF than CRUNCEP. Small differences in the simulated presence of permafrost (or the number of sites ISPF) are to be expected given the possibility of errors in the meteorological forcing and local variations in site-level characteristics, but large differences can indicate problems with the model setup and parameterizations.

### 3.2 Initial model performance

The *Base model* experiment simulates a permafrost area (PA) of 8.6 Mkm$^2$ (north of 60° S; Table 2) with permafrost confined to northern Siberia, Alaska and the northern edge of Canada (Figure 2). This low PA is in line with that simulated by CLASS-CTEM when coupled within the CanESM, although the spatial distribution is different due to the different atmospheric forcing (Koven et al., 2013). Also plotted in Figure 2 is the PE estimate of Brown et al. (1997). The Brown et al. (1997) dataset gives permafrost spatial distribution in four classifications which are not directly comparable to ALTs but may be used to give a general indication of PA from an independent estimate. Owing to the coarseness of the model grid CLASS-CTEM is not able to simulate isolated or sporadic permafrost. For regions of discontinuous and continuous permafrost, comparing the estimated distribution of Brown et al. (1997) to the modelled ALT indicates poor agreement.

With such a small permafrost area many of the GTN-P ALT sites were ISPF as mentioned above. Of the GTN-P ALT sites where CLASS-CTEM simulated permafrost, the *Base model* simulations show overly shallow ALTs with an average mean absolute error (MAE; described in Section 2.3) of 0.410 m. Thus it appears the modelled soil temperatures are too warm in the more southerly permafrost domain (PD), leading to no permafrost simulated, and too cool at the higher latitudes. However, it should be noted that the model configuration of three ground layers in this experiment makes an accurate estimation of the ALT difficult since the lowest model layer is quite thick (3.75 m).

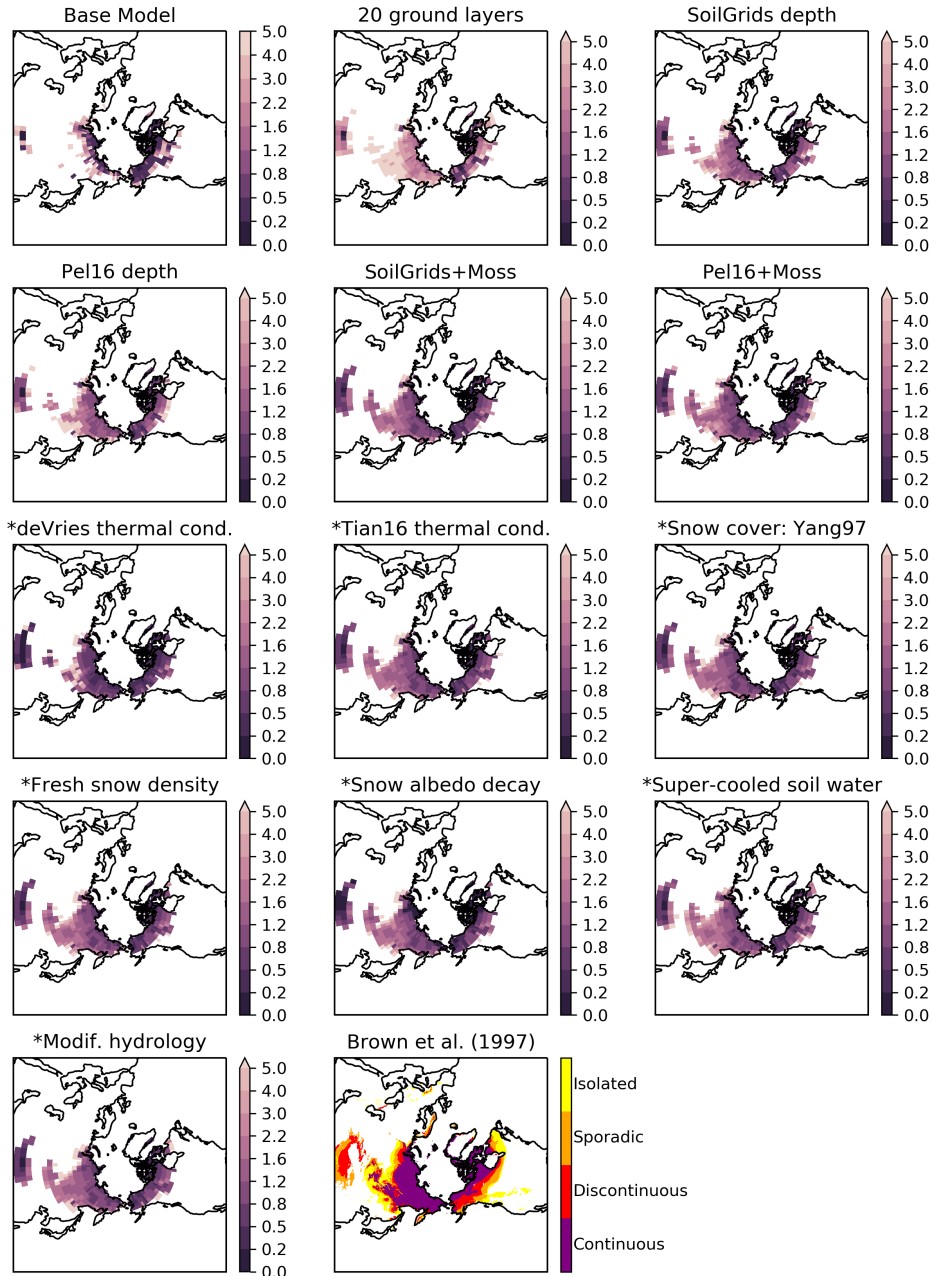

**Figure 2.** ALTs in meters for experiments listed in Table 1 alongside the permafrost map of Brown et al. (1997) (bottom right). Experiments with an asterisk prefixing their name use a model configuration based on the *SoilGrids+Moss* setup. All experiments shown here use CRUNCEP for the meteorological forcing.

**Table 2.** Permafrost area as simulated by CLASS-CTEM (average of 1996 - 2015) along with literature estimates for terrestrial permafrost north of 60°S. The number of GTN-P sites which CLASS-CTEM incorrectly simulated as permafrost free (ISPF) is also listed along with the number of corresponding grid cells in square brackets. These GTN-P sites were removed from further analysis since ALT is not defined in locations with no permafrost. The numbers in parentheses indicate the values when CRUJRA55 was used as the meteorological forcing instead of CRUNCEP. See Section 2.3.4 for distinction between permafrost area and permafrost region.

| Experiment | Permafrost Area ($10^6$ km$^2$) | Number of sites [grid cells] ISPF |
|---|---|---|
| *Base model* | 8.6 | 15 [8] |
| *20 ground layers* | 16.7 | 7 [5] |
| *SoilGrids depth* | 15.7 | 8 [6] |
| *Pel16 depth* | 15.7 | 8 [6] |
| *SoilGrids+Moss* | 17.9 (19.8) | 7 (6) [5 (4)] |
| *Pel16+Moss* | 18.5 (19.8) | 7 (6) [5 (4)] |
| *deVries thermal cond.* | 16.2 (17.8) | 8 (6)[6 (4)] |
| *Tian16 thermal cond.* | 21.2 | 6 [4] |
| *Snow cover: Yang97* | 19.3 (20.8) | 6 (6) [4 (4)] |
| *Snow cover: Brown03* | 19.0 | 6 [4] |
| *Fresh snow density* | 18.9 | 6 [4] |
| *Snow albedo decay* | 15.6 | 6 [4] |
| *Super-cooled water* | 20.1 | 6 [4] |
| *Modif. hydrology* | 19.5 | 6 [4] |
| Literature estimates | Permafrost Area ($10^6$ km$^2$) | |
| Zhang et al. (2000) | 12.2 - 17.0 | |
| Gruber (2012) | 12.9 - 17.7 | |
| Literature estimates | Permafrost Region ($10^6$ km$^2$) | |
| Zhang et al. (1999) | 22.8 | |
| Gruber (2012) | 21.7 with a range of 18.7 to 24.3 | |

### 3.3 Increasing the number of ground layers

Increasing the number of ground layers from 3 to 20 decreases the number of GTN-P ALT sites ISPF from 15 to 7 (Table 2). Figure 3 shows the difference between the simulated and observed ALT at each grid cell with GTN-P ALT sites for selected experiments. The average MAE computed against the GTN-P ALT observations for *Exp 20 ground layers* is over 2.5 m with simulated ALTs strongly overestimated (Figure 3). When the number and depth of ground layers is increased, but the soil permeable depth is left unchanged, CLASS-CTEM simulates the ground layers below the permeable soil depth as impermeable

bedrock. The absence of water and therefore of heat consumption by melting ice in these lower ground layers causes the model soil column to be generally too warm. However, the total global PA increases from 8.6 Mkm$^2$ simulated by the *Base model* to 16.8 Mkm$^2$ (Table 2) with an increase in permafrost area primarily in the southern fringes of eastern Siberia and Canada along with a general deepening of ALT across the high latitudes (Figure 2). This seeming incongruity of warmer soils with a larger permafrost area likely relates to moving the boundary of zero heat flux from 4.1 m, a depth where seasonal temperature variations can penetrate, to 61.4 m. The shallower modelled soil column in the *Base model* inhibits the formation of permafrost because of the concentration of the annual heat flux oscillation in the upper few meters of the soil.

The wMAE calculated for each season from CLASS-CTEM's simulated ground temperatures compared to GTN-P borehole temperatures for three depth zones shows an improvement at all depths and seasons for *20 layers* over *Base model* (Figure 4). Generally, across all experiments, CLASS-CTEM performs better with increasing depth. Seasonally, winter is generally simulated best with summer showing the highest wMAE values. These patterns indicate that the largest challenges to accurate ground temperature simulation are coming from the high variability in forcing at the land surface and from the difficulty in accurately simulating the summertime heat pulse into the ground column.

To look in closer detail at the model performance for the GTN-P borehole sites, Figure 5 shows the Gaussian kernel density estimate (KDE) derived from differences between the simulated and observed borehole temperatures. For shallow soils, as the seasons progress from winter to fall, the proportion of instances with a strong cold bias decreases with a warm soil bias taking over in summer, especially in the shallowest depth band. This would indicate the modelled soil heat fluxes are somewhat exaggerated. The fall period generally has the least bias, potentially due to the loss of the warm summer bias but prior to the establishment of the cold winter bias.

### 3.4 Increasing the soil permeable depths

Changing the soil permeable depth dataset to SoilGrids (Exp. *SoilGrids depth*) from Zobler86 gives a general improvement over the *20 ground layers* simulations with a drop in average MAE to 1.162 m at the GTN-P ALT sites (Figure 3). There is also a shift to shallower ALTs (Figure 2) with a slight decrease in PA to 15.7 Mkm$^2$, which is within the range literature estimates (Table 2 and discussed further in Section 2.3.4). The greater permeable depths associated with SoilGrids lead to deeper penetration of water into the soil, resulting in more water being allocated to runoff than made available for plant transpiration or soil evaporation (Supplement Figure 3). Simulations with the alternative soil permeable depth dataset (*Pel16 depth*) generally show similar patterns of latent heat flux, runoff and LAI (not shown) to the *SoilGrids depth* experiment. The *Pel16 depth* simulations have better agreement with the GTN-P ALT observations reducing the wMAE to 0.757 m (Figure 3). *SoilGrids* also further improves the model's performance at all depths and seasons compared to the GTN-P borehole sites (Figure 4).

Numerous studies have pointed to the importance of increasing the simulated ground column depth and number of ground layers to better capture the decay with depth of the influence of multi-decadal variability (e.g. Smerdon and Stieglitz, 2006; Alexeev et al., 2007; Nicolsky et al., 2007; Paquin and Sushama, 2014). Of particular relevance to our study, Paquin and Sushama (2014) used CLASS in CRCM5 and found shallow soil configurations (permeable depth < 1 m throughout much of

the model domain) to lead to overly strong seasonal cycles with resulting overly deep ALTs, similar to the work of Smerdon and Stieglitz (2006), and in line with our *Base model* simulation with its small estimated PA.

The availability of comprehensive global soil permeable depth datasets is relatively recent. Previous studies would often assume a constant permeable soil depth, either shallow (Dankers et al., 2011) or deep (Lawrence et al., 2008) with the deeper

layers hydrologically inactive. Comparing the three permeable depth datasets (Zobler86, SoilGrids, and Pel16; Supplement Figure 1) shows Zobler86 to be by far the shallowest while SoilGrids and Pel16 disagree on the spatial distribution of the permeable depths for the high latitude regions. Pel16 shows deep soils in the Canadian boreal forest, Finland and central southern Russia with shallower soils in the Siberian plateau. SoilGrids has more very deep soils (>50 m) especially in the West Siberian region and the Urals. These differences in permeable depth have an impact on the simulated ALT as the SoilGrids

and Pel16 experiments perform quite differently at the GTN-P ALT sites (Figure 3) due to the strong impact of freezing and thawing of water in the soil column.

### 3.5 Adding an upper layer of organic matter/moss to the soil column

CLASS-CTEM ALTs with both Pel16 and SoilGrids are generally biased deeper than observed at the GTN-P sites (Figure 3) indicating that the ground surface is either overly insulated from the cold atmosphere during the winter or is absorbing too much

heat during the summer months. The principal modulating influences on ground heat fluxes in cold regions are hydrology, snow cover (both of which we deal with later), vegetation structure and function, and topography (Loranty et al., 2018). Vegetation canopies shade the soil surface, attenuating radiation and reducing warming in the summer season. As well dense forests capture snow in the canopy which prevents it from reaching the ground and insulating the soil surface further cooling soils. Another aspect of vegetation influence is the insulating effect of a surface layer of moss or organic matter. Mosses are generally

more abundant at high latitudes and have been shown to decrease growing season surface soil temperatures (Turetsky et al., 2012). The effect of mosses on the ground heat flux has also been demonstrated through field experiments (Gornall et al., 2007; Van Der Wal and Brooker, 2004) and modelling studies have incorporated organic layers (e.g. Lawrence et al., 2008; Paquin and Sushama, 2014) or bryophytes (Porada et al., 2016) to improve permafrost dynamics. Exps. *SoilGrids+Moss* and *Pel16+Moss* both incorporate a non-photosynthetic moss layer in place of the first layer of soil (see Section 2.2) and both

simulate generally shallower ALTs than their parent simulations (*SoilGrids depth* and *Pel16 depth*, respectively; Figure 2). The effect of moss introduction for the *SoilGrids+Moss* experiment is to reduce average MAE from 1.162 to 0.472 m for the GTN-P ALT sites (Figure 3) The general cooling influence is evident by comparing to the GTN-P ALT sites (Figure 3) and also through the increase in simulated PA from 15.7 to 17.9 Mkm$^2$. A similar improvement is seen for Exp *Pel16+Moss*. The high porosity of the moss layer causes less water to be available at the surface for evaporation, reducing the latent heat flux and

making more water available for runoff, and its insulating effect keeps the soil surface cooler, which reduces plant growth and also the sensible heat flux (Supplement Figure 4). The reduction in plant growth due to cooler soils also reduces water uptake for transpiration further increasing runoff.

Comparing simulated ground temperatures to observations at the GTN-P borehole sites shows a slight increase in wMAE at all depth ranges and seasons compared to the *SoilGrids* simulation (Figure 4). Comparing the KDE plots of the bias distri-

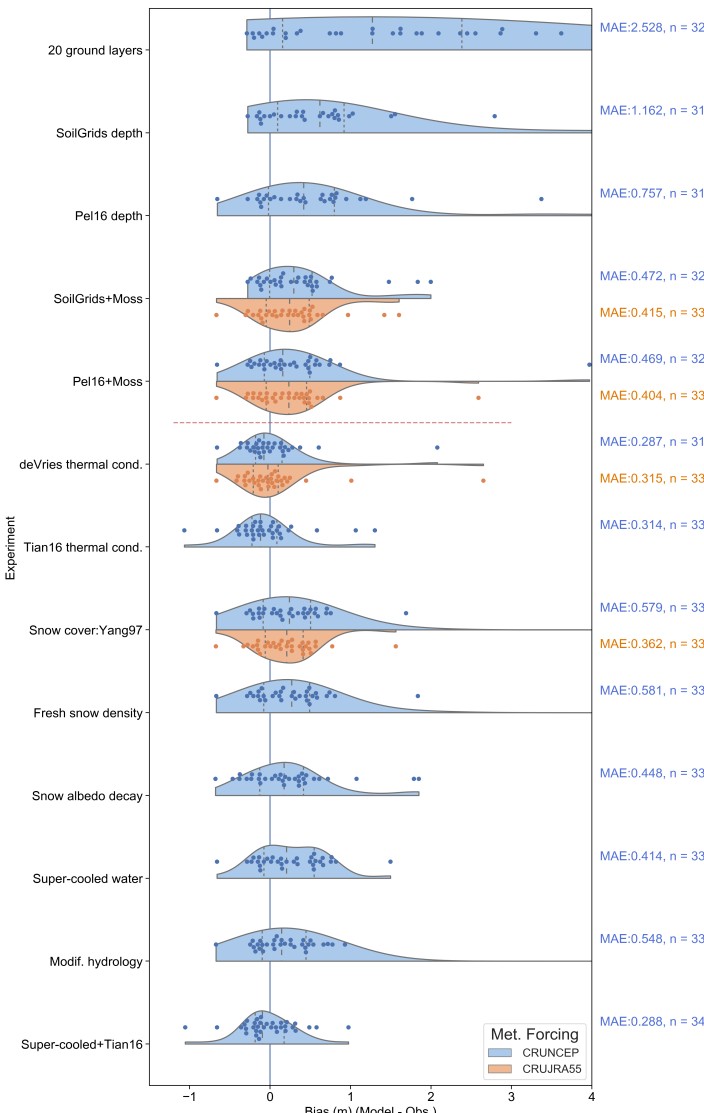

**Figure 3.** Differences between the ALTs from the experimental model runs and those of the Global Terrestrial Network for Permafrost ALT sites (Supplement Table 1). Each dot represents a grid cell with one or more GTN-P sites (see Section 2.3). In this representation (a 'bee swarm'), displacement in the y-direction is only to allow each data point to be visible. The background shading is a Gaussian kernel-density estimate (KDE) with the quartiles of the distribution indicated by dashed vertical lines within the KDE plot. The mean absolute error (MAE) is produced by calculating the MAE at each grid cell and taking the average across all cells. As the number of sites ISPF differs between experiments (Table 2) the number of grid cells where CLASS-CTEM simulated permafrost is also listed. The total number of grid cells with GTN-P sites is 37. The two meteorological forcings are shown for the experiments where the CRUJRA55 forcing was also used. Experiments below the dashed red line use the model setup from Exp. *SoilGrids + Moss* as their starting point (Table 1).

bution between modelled and observed borehole temperatures for the *SoilGrids+Moss* and the *SoilGrids* simulations shows an increased cool bias in the shallow soil which is especially evident in summer (Figure 5). This bias extends deeper into the soil column, albeit weakening with depth. The cooling of soils due to the incorporation of a moss layer was also found by Porada et al. (2016), however their simulations included a dynamic extent for moss cover. The creation of a cold bias due to the introduction of a moss layer is reasonable considering that the moss layer was applied to all areas uniformly. While this experiment was intended to understand the impact of moss on simulated ground temperatures, future work should attempt to place moss with a more realistic distribution, similar to Porada et al. (2016).

Comparing the model experiment outputs to the GTN-P sites in Figure 3 it is evident that increasing the number of ground layers and the soil permeable depth and incorporating a top layer of moss/organic matter improves the simulated ALTs. These changes have been suggested by other studies as mentioned above and our results are in line with them. The next experiments use the model configuration from *SoilGrids+Moss* as a starting point. While Pel16 generally gave better average MAE values than SoilGrids for ALT compared to the GTN-P sites (Figure 3), SoilGrids appears to be better validated (c.f. Shangguan et al., 2017, Figures 9 - 11). Both datasets, however, suffer from sparse data in high latitudes (e.g., Shangguan et al., 2017, Figure 2). Additionally, while it appears that the addition of moss can introduce a summer cool bias in ground temperatures (as discussed above), given the extensive distribution of bryophytes (c.f. the simulated distribution in Figure 4b in Porada et al., 2016), we chose to include moss in our further simulations.

## 3.6 Testing alternate soil thermal conductivity formulations

Exp. *Tian16 thermal cond.* tests the Tian et al. (2016) formulation, which is based on de Vries (1963) but explicitly accounts for the influence of ice (see Section A2 and the Supplement section 1). The new formulation simulates a much larger PE than *SoilGrids+Moss* at 21.2 Mkm$^2$ with generally shallower ALTs in most regions except for the western edge of simulated Siberian permafrost (Supplement Figure 5). The average MAE at the GTN-P ALT sites is reduced to 0.314 m (Figure 3) however, at the GTN-P borehole sites, the simulated ground temperatures are biased cold, primarily in summer and fall, and worsening with depth (Figure 4 and 5).

## 3.7 Changing the relationship between snow depth and snow cover

Two experiments investigated different relationships between snow depth and the grid cell snow cover in CLASS-CTEM (*Snow cover: Yang97* and *Snow cover: Brown03*). These modifications increased global PA (~1.2 Mkm$^2$) with a slightly higher PA estimated for *Snow cover: Yang97* (Table 2). For the GTN-P ALT sites, both snow cover experiments increased average MAE from 0.472 m for *SoilGrids+Moss* to 0.579 m and 0.622 m for *Snow cover: Yang97* and *Snow cover: Brown03*, respectively. Comparing the simulated SWE from both *Snow cover: Yang97* and *Snow cover: Brown03* to Blended-5 (see Section 2.3) shows a slight improvement in model performance compared to both *Base model* and *SoilGrids+Moss* throughout the snow year, which tends to be more pronounced during fall and winter (Supplement Figure 6) although there is little difference between the two snow cover experiments.

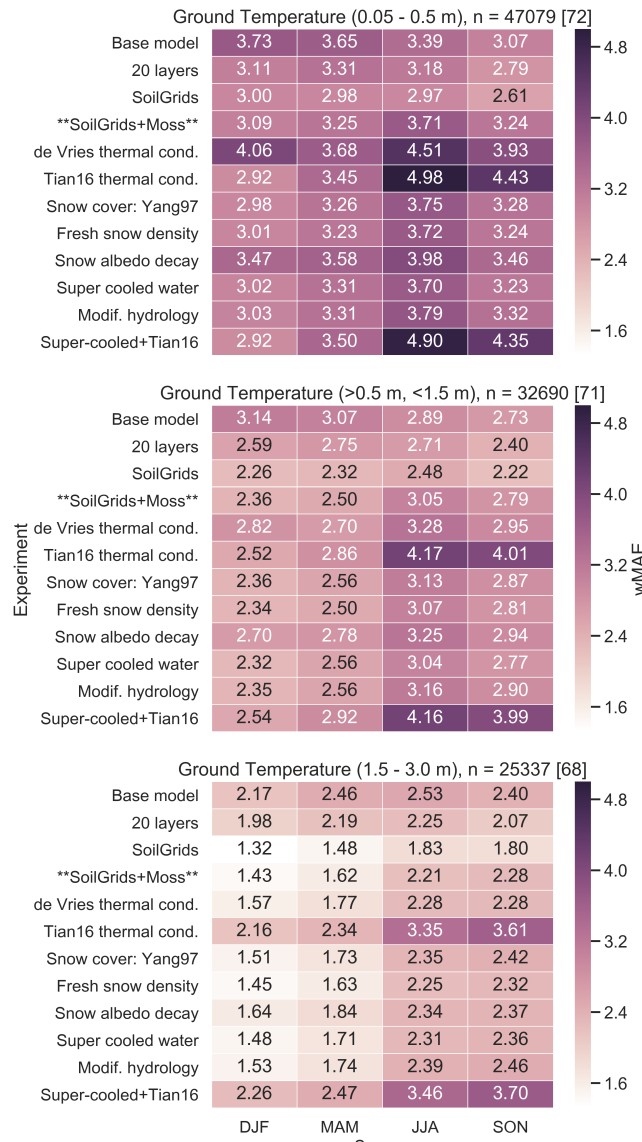

**Figure 4.** Weighted mean absolute error (wMAE, °C) between the simulated ground temperatures and those of the GTN-P borehole temperature sites (Supplement Table 2) for three depths: 0.05 - 0.5 m, 0.5 m - 1.5 m, and 1.5 - 3.0 m. The weighted mean absolute error (wMAE) is produced by calculating the wMAE for each depth range and season at each site within a grid cell and taking the average across all grid cells (see Section 2.3).The number of observations differs between depths and is listed along with the number of CLASS-CTEM grid cells with GTN-P borehole sites in square brackets. The colour of the text annotations is purely for clarity. The wMAE of CRUNCEP surface air temperatures compared to air temperatures measured at the GTN-P sites is 2.17 °C, 2.46 °C, 2.53 °C, and 2.40 °C for DJF, MAM, JJA, and SON, respectively over 25 337 monthly observations

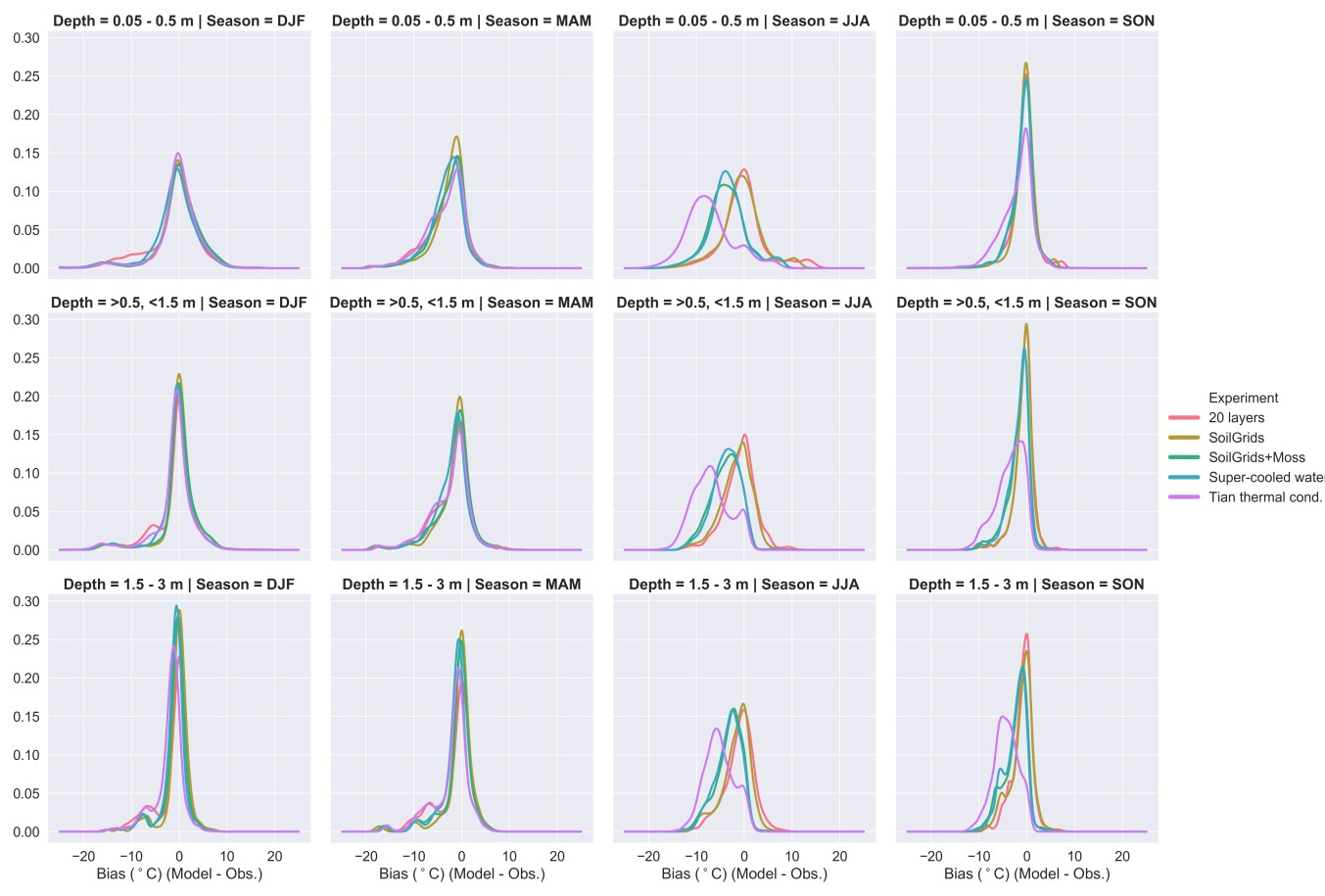

**Figure 5.** Gaussian kernel density estimates for the difference between the simulated ground temperatures and those of the GTN-P borehole temperature sites (Supplement Table 2) for three depths: 0.05 - 0.5 m, 0.5 m - 1.5 m, and 1.5 - 3.0 m, for each season and for selected experiments. The bandwidth was chosen using Scott's rule of thumb (Scott, 1992).

Changes in snow cover can lead to large changes in albedo due to the significant brightness difference between snow and vegetation/bare ground. To investigate the impact of these experiments on albedo we evaluated seasonal averages of simulated albedo against MODIS observations over latitudes northward of 45°N for the period 2000 to 2013. We find the spring (AMJ) albedo from the various simulations is about the same (Supplement Fig. 7).

5  **3.8  Considering wind speed in the calculation of fresh snow density**

In CLASS-CTEM, the density of freshly fallen snow depends on the ambient air temperature (Eqn. A19). Exp. *Fresh snow density* tested a parameterization from the CROCUS model that also includes wind speed in this calculation (Eqn. A20) which yielded an increase in PA to 18.9 Mkm$^2$. Compared to the GTN-P ALT sites, the *Fresh snow density* results are similar to those

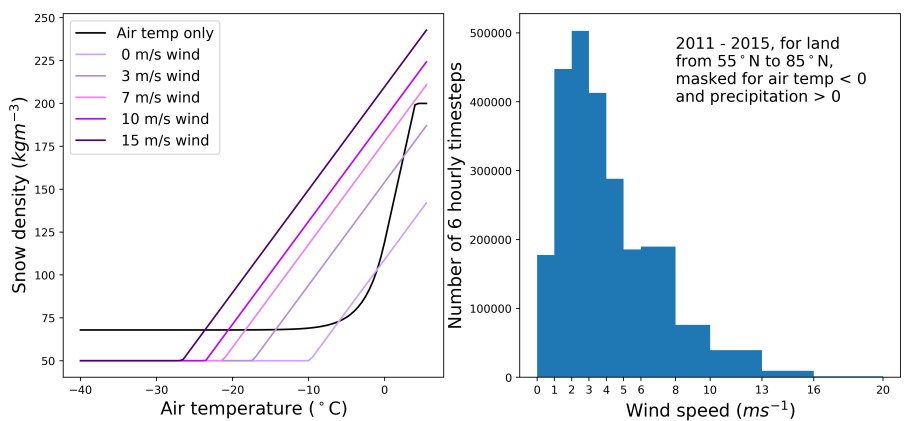

**Figure 6.** Left: Snow density as a function of air temperature for the original CLASS-CTEM formulation (Eqn A19) and for Exp. *Fresh snow density* which includes consideration of wind speed (Eqn A20 -purple lines indicate different wind speeds). Right: Histogram of wind speeds for the period 2011 to 2015 from the CRUNCEP meteorological dataset.

of the snow cover experiments with no improvement in average MAE (0.581 m; Figure 3) and no discernible impact upon modelled DJF SWE compared to Blended-5 or upon spring (AMJ) albedo compared to MODIS (Supplement Figure 6 and 7).

The typical wind speed in the CRUNCEP meteorological forcing dataset when snow is falling is in the range of 1 - 5 m s$^{-1}$ (Figure 6). With Eqn A20, the density of freshly fallen snow tends to be lower at very low wind speeds then higher as wind
speed increases for the same air temperature. The generally higher density of fresh snow with the CROCUS parameterization results in a snow pack with higher thermal conductivity (Sturm et al., 1997) and thus cooler soils as evident from the expansion in PA for the *Fresh snow density* experiment (Figure 2). Both the original CLASS-CTEM parameterization and that of the CROCUS model produce fresh snow densities within the range of observations. Roebber et al. (2003) evaluated 1650 snowfall events from 28 continental US sites and found the density of freshly-fallen snow to vary from 21.4 to 526.3 kg m$^{-3}$ with a
median value of 70.9 kg m$^{-3}$ (for snowfall events where the wind speed was $\leq$9 m s$^{-1}$).

### 3.9 Adopting an efficient spectral method for snow albedo decay

Changing the snow albedo decay parameterization from an exponential form (Verseghy, 2017) to an efficient spectral parameterization (Dickinson, 1983)(Exp *Snow albedo decay*) slightly improves average MAE at the GTN-P ALT sites (Figure 3) while decreasing PA (15.6 Mkm$^2$), reflecting a near uniform deepening of ALT with the exception of small areas on the west-
15 ern edge of the Siberian PD (Figure 2; Supplement Figure 5; Table 2). The efficient spectral method for albedo decay generally produces lower albedos than CLASS-CTEM's original exponential parameterization. The impact upon spring albedo and SWE leads to a notable decline in model performance compared to observation-based datasets (Supplement Figures 6 and 7).The CRUJRA55-forced experiments, on the other hand, give slightly better spring albedo for all experiments forced with that meteorological dataset. This could be due to the sub-monthly variability difference of CRUJRA55 compared to CRUNCEP as Beer

et al. (2018) found one of the largest impacts of changing climate variability in model forcing to be snow depth. The lower albedo in the *Snow albedo decay* experiment leads to a smaller snowpack which melts earlier resulting in reduced spring runoff, a longer growing season, and a higher LAI. The warmer land surface results in larger ALTs. At the GTN-P borehole sites, the *Snow albedo decay* experiment's warmer ground layers gives a noticeable increase in wMAE values across all seasons and most depth bands.

### 3.10 Allowing unfrozen water in frozen soils

The inclusion of unfrozen water in frozen soils (Exp. *Super cooled water*) increased PA to 20.1 Mkm$^2$ with a minor improvement at the GTN-P ALT sites (Figure 3). The GTN-P borehole sites showed little change in the wMAE values (Figure 4). The larger PA for this experiment could be reflecting the thermal conductivity differences between completely frozen soil and frozen soil with some residual liquid water. The differences in bulk thermal conductivity would slow heat transfer into the deeper ground layers for the *Super cooled water* simulation during periods where the soil layer temperature is below 0°C. As a result spring warming would be slower to reach deeper layers.

Ganji et al. (2015) investigated streamflow for 21 watersheds in eastern Canada using CLASS and the WATROUTE routing scheme. They report their modifications (super-cooled soil water, fractional permeable area and modified hydrology due to ice; discussed in Section A7) improved streamflows particularly during the spring melt. The changes were attributed to reduced hydraulic conductivity of frozen soils causing more snow melt runoff and less infiltration. We did a rudimentary comparison of our simulated seasonal runoff for seven major river basins that drain permafrost regions (Figure 7; Section 2.3). As the CLASS-CTEM simulations did not include excess ground ice (e.g. slab ice such as ice wedges or lenses commonly found in regions affected by thermokarst processes), groundwater or interflow, all of which could increase runoff (baseflow) in the summer and fall seasons, we limit our discussion to the spring and winter seasons. The *Super cooled water* experiment has lower spring runoff than both *Base model* and *SoilGrids+Moss* but higher winter runoff, making it more in line with observed river discharges (Figure 7).

Given the *Super cooled water* and *Tian16 thermal cond.* simulations had the lowest average MAE at the GTN-P ALT sites (Figure 3) a simulation was run with both of these parameterizations included (Exp. *Super-cooled+Tian16*). This experiment further reduced the average ALT MAE but considerably worsened simulated ground temperatures at the GTN-P borehole sites (Figure 4). This incongruity between model performance at the ALT and borehole sites could be reflecting biases due to the spatial distribution of the sites (see Figure 1), the differing number of observations of ALT vs. borehole temperatures, or to biases in the observations themselves, which is discussed in Section 3.13.

### 3.11 Modifying hydrology due to ice

The *Modif. hydrology* experiment modified soil matric potential and saturated hydraulic conductivity to account for the impact of frozen water following the work of Farouki (1981) and Koren et al. (1999). These changes yielded a simulated PA of 19.5 Mkm$^2$ (Table 2) with generally slightly deeper ALTs in much of the high latitude PD compared to *SoilGrids+Moss* (Figure 2 and Supplement Figure 5) and poorer average MAE for the GTN-P ALT sites Performance at the GTN-P borehole

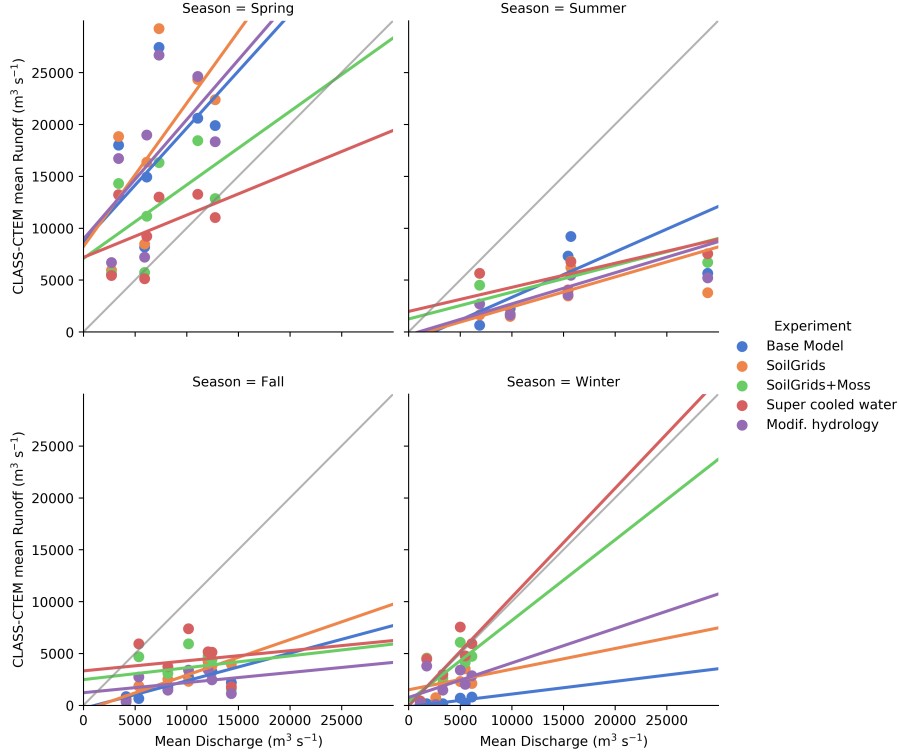

**Figure 7.** Mean 1965 - 1984 seasonal discharges of major rivers draining permafrost regions (Ob, Volga, Lena, Yenisei, Yukon, Mackenzie and Amur; UNESCO Press, 1993) compared to total runoff from selected model runs for the same period. Each dot represents one river basin. The CLASS-CTEM simulated runoff is not routed thus only seasonal values are compared.

sites is similar to Exp. *SoilGrids+Moss* (Figure 4). Since the modifications to soil matric potential and saturated hydraulic conductivity (Equations A35 and A36) generally decrease water mobility in soils with ice present, the *Modif. hydrology* soils are generally wetter, allowing higher annual latent heat flux and supporting higher LAI. The *Modif. hydrology* experiment has similar runoff to the *Base model* experiment with higher spring runoff than observed river discharges while the winter
5   runoff is reduced compared to *SoilGrids+Moss* and is also smaller than the observed river discharges (Figure 7). To investigate synergistic effects between the two modifications (*Modif. hydrology* and *Super cooled water*), a simulation was run with both modifications applied (similar to Ganji et al.'s Exp. 3). This simulation gave slightly higher spring runoff but similar winter runoff compared to *Modif. hydrology* (not shown). Thus it appears, with respect to runoff, the modifications to hydrology have a stronger influence than super cooled soil water, in line with the conclusion of Ganji et al. (2015) that the primary effect is to
10   reduce hydraulic conductivity which decreases infiltration and increases snow melt runoff.

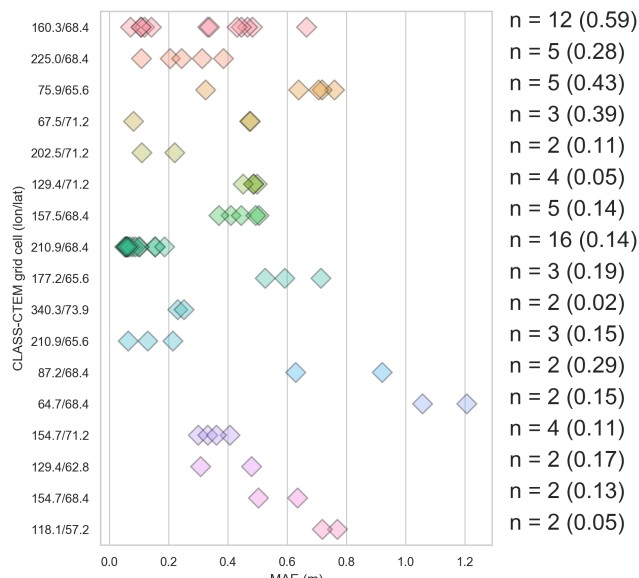

**Figure 8.** Mean absolute error (MAE) for CLASS-CTEM grid cells with multiple GTN-P ALT sites for the *SoilGrids+Moss* simulation. The number of ALT sites is listed along with the range in MAE in each grid cell in parentheses.

### 3.12 Influence of sub-grid heterogeneity

The CLASS-CTEM model grid used in our study is the same as that used in the CanESM. From the experiments conducted the lowest average MAE at the GTN-P ALT sites we are able to achieve is about 0.4 m. With the size of our model grid cells, what is the best MAE we can reasonably expect given the sub-grid heterogeneity at the observation sites? Many of the GTN-P
ALT measurements are performed on an 11 x 11 sampling grid covering between 1 km$^2$ and 1 ha giving 121 data points at one point in time per site; the mean standard deviation of measured ALT over these sampling grids varies from 0.02 m to 0.49 m (Supplement Table 1). However, one square kilometre is still small compared to model grids ranging in size from hundreds to thousands of km$^2$. One measure of the influence of sub-grid heterogeneity can be obtained by considering the MAE per site in the grid cells where we have more than one GTN-P ALT site (Figure 8). For these grid cells, the spread in MAE at each
site ranges from 0.01 m (grid cell with 2 sites) to 0.59 m (12 sites). While it is not reasonable to directly compare the sub-grid range of MAE to the model average MAE shown in Figure 3, Figure 8 demonstrates that sub-grid heterogeneity is a significant source of variability in ALT within model grid cells and that variability will impose constraints on the lower limit of MAE that is attainable by the model.

For the GTN-P borehole sites, the wMAE in temperature bias for the model varies between ca. 1.5 and 3.7 °C (Figure
4), depending on depth and season. As with ALT, what is a reasonable wMAE for ground temperatures given the size of the model grid cells and the discrete nature of a borehole? To better understand the role of sub-grid heterogeneity in borehole temperatures, we make use of the Slave Province Surficial Materials and Permafrost Study (SPSMPS; Gruber et al., 2018).

The SPSMPS collected air and ground temperature measurements for 15 m x 15 m plots with hourly borehole temperatures at thirty-five boreholes all located within a ca. 1200 km$^2$ area. The observed screen-level temperatures are generally reasonably close to those of CRUNCEP but CRUNCEP has slightly cooler summer temperatures (Supplement Figure 8). What is most striking about the borehole temperatures at Lac de Gras is the large spread in ground temperatures at all depths and in most seasons (Figure 9). The temperature range is smallest in fall and spring when the soils are thawing or freezing and largest in winter with differences varying from 12 to over 20 °C depending on the soil depth. This remarkable spread in temperature is due to variations in slope, aspect, soil moisture, soil texture, soil organic matter content, and vegetation type and distribution. The simulated ground temperatures from two experiments are plotted alongside the boreholes (*SoilGrids* and *SoilGrids+Moss*). As the model is driven by CRUNCEP and we have no precipitation information for the SPSMPS sites, it is difficult to determine the cause of any biases. Also, although the SPSMPS sampling area is considerably larger than the GTN-P sites, the same arguments apply concerning the mismatch of scales between the observational area and the model grid, and the variability introduced by sub-grid heterogeneity.

An additional measure of how reasonable the model wMAE is at the borehole sites can be obtained by comparing the CRUNCEP screen-level temperature, which is used to force the model, and the observed screen-level temperature at each GTN-P site. The MAE for screen-level temperature is between 2.17 and 2.53 °C across all seasons. Therefore the model's wMAE range for shallow soil of ca. 3 to 3.7 °C varies from ca. 0.8 to 1.2 °C above that of the MAE for CRUNCEP's screen-level temperature (for the *SoilGrids+Moss* simulation). Given the large spread in borehole temperatures in a relatively small area at the SPSMPS sites, and the MAE of the model's forcing air temperature, it appears the model's wMAE can be considered reasonable.

## 3.13 Influence of bias due to ALT or borehole sampling locations

Temperature in individual boreholes and ALT at individual sites often differ from the grid cell they are compared with because of sub-grid variability as discussed above. The underlying spatial variation of ground temperature, even at distances smaller than 1km is well documented (Smith, 1975; Morse et al., 2012; Gubler et al., 2011). If the locations of GTN-P sites were randomly sampled, sub-grid effects would be expected to cancel out and, consequentially, a mean bias (cf. Figure 3) close to zero would be indicative of good model performance. In reality, however, the choice of GTN-P measurement locations are likely biased and the nature and consequences of this bias are difficult to assess. For example, ALT sites are likely to be biased toward fine-grained and organic-rich soils and locations with small ALT where probing can be carried out. The choice of ALT and borehole sites in areas of sporadic permafrost is likely to be biased towards cold areas in the landscape. This is because ALT requires permafrost and because permafrost researchers are unlikely to drill, instrument and operate boreholes in seasonally frozen ground. Finer-scale local studies have noted that observations are strongly biased towards permafrost existence (Boeckli et al., 2012). The melt of excess ice from the top of permafrost presents an additional source of bias that may result in ALT data showing values of seasonal thaw depth that underestimate the amount of ground ice that was melted due to frost-table probing without recording surface subsidence (Shiklomanov et al., 2013). In summary, it is likely that a slightly positive model bias,

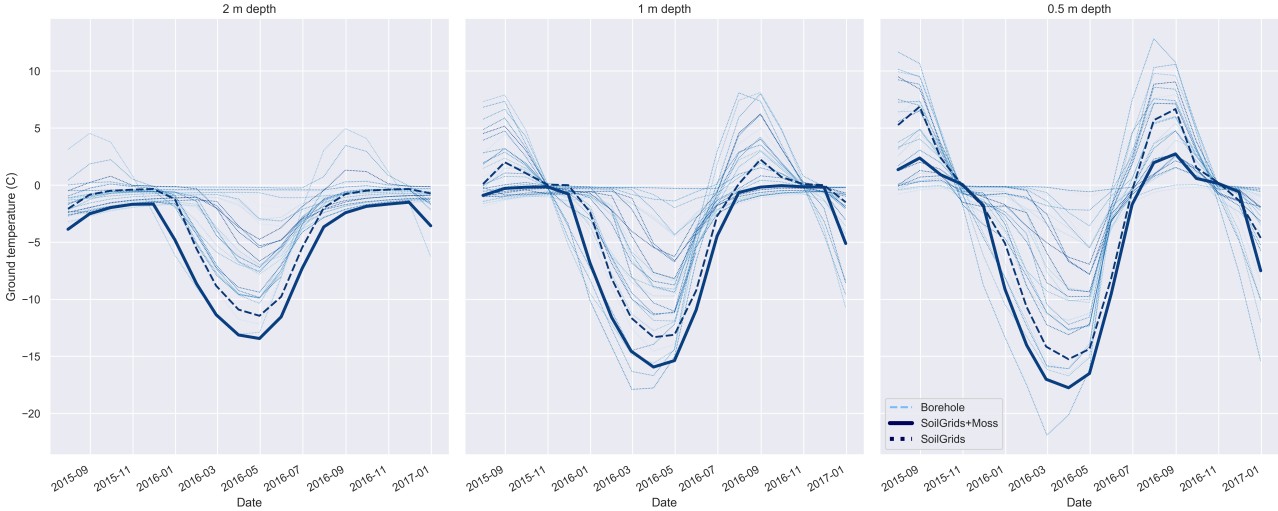

**Figure 9.** Borehole temperatures for 0.5, 1, and 2 m depths from the Slave Province surficial materials and permafrost study (SPSMPS, Lac de Gras region, NWT, Canada; Gruber et al. (2018)) along with CLASS-CTEM simulated ground temperatures for Exp. *SoilGrids* and *SoilGrids+Moss*. The thirty-five boreholes are each represented by a single line and are all located within a ca. 1200 km$^2$ area. The model output is from the grid cell corresponding to the SPSMPS study area.

i.e. higher temperatures and greater ALT simulated than observed, would correspond to a model that best represents reality. Quantifying that effect however is beyond the present study.

## 4   Conclusions

The performance of CLASS-CTEM in cold regions has been investigated in the past by numerous researchers who have suggested several modifications to improve the model's performance in these regions. Drawing from these recommendations and other studies, 18 experiments were carried out to investigate the influence of: 1) the number of ground layers, 2) soil permeable depth datasets, 3) the addition of a moss layer, 4) changing the soil thermal conductivity formulation, 5) altering the derivation of snow cover based on snow depth, 6) adding the effect of wind speed to the calculation of fresh snow density, 7) changing the model's snow albedo decay calculation to an efficient spectral parameterization, and 8) modifications to frozen soil hydrology including allowing unfrozen water in frozen soils and an alteration to hydraulic conductivity and soil matric potential for the presence of ice. Two soil permeable depth datasets were tested (Pelletier et al. (2016) and SoilGrids; Shangguan et al. (2017)) along with two meteorological datasets (CRUNCEP v.8; Viovy (2016) and CRUJRA55 v.1.0.5; Kobayashi et al. (2015); Harris et al. (2014)). The simulated active layer thicknesses (ALTs) were compared to 1570 observations from 97 sites from the Global Terrestrial Network for Permafrost (GTN-P; Supplement Table 1, Figure 1), the simulated soil temperatures to 105 106 monthly observations at 132 GTN-P borehole temperature sites (Supplement Table 2), 35 borehole sites from the Slave Province Surficial Materials and Permafrost Study (SPSMPS; Gruber et al., 2018), surface albedo to a remotely-sensed

dataset (MODIS MCD43C3), snow water equivalent (SWE) to a blend of five observation-based datasets (Blended5; Mudryk et al., 2015), and seasonal runoff to river discharges for major rivers draining the Arctic (UNESCO Press, 1993) as well as literature estimates of permafrost area (Table 2).

The original model version had an overly small simulated permafrost area of 8.6 Mkm$^2$ which was almost doubled to 16.7 Mkm$^2$ by increasing the number and depth of ground layers. Of the two soil permeable depth datasets, Pelletier et al. (2016) gave consistently lower average mean absolute errors (MAE) at the GTN-P ALT sites compared to SoilGrids. However, SoilsGrids was chosen for further simulations as this dataset appears to be better validated (Shangguan et al., 2017). For the two meteorological datasets used, the permafrost specific results depended on the model configuration and parameterizations tested. More consistently, spring albedo appeared to be better simulated using CRUJRA55 while winter SWE was slightly better with CRUNCEP. Changes to the model configuration by increasing soil permeable depths using the SoilGrids dataset, and adding a layer of moss reduced the average MAE at the GTN-P ALT sites from over 2.5 m (Exp. *20 ground layers*) to 0.472 m (Exp. *SoilGrids+Moss*). While most alternate parameterizations either degraded model performance at the GTN-P ALT and borehole sites, or degraded the performance of another model output such as albedo or SWE, incorporating unfrozen water in frozen soils following Niu and Yang (2006) is being considered for inclusion in future versions of CLASS-CTEM. A simulation with the Niu and Yang (2006) parameterization resulted in an average MAE of 0.414 m at the GTN-P ALT sites, relatively small impacts on wMAE at the GTN-P borehole sites, and a possible improvement in seasonal runoff. Further assessment of the improvements in runoff using a river routing scheme are needed before this parameterization will be fully adopted. Based on the tests performed here, the optimal model configuration will include more ground layers to a greater depth, soil permeable depths from the SoilGrids dataset, and moss in locations where it is appropriate. These changes give a simulated permafrost area of between 15.7 to 17.9 Mkm$^2$ (Table 2) which reasonably close to the expected 15 Mkm$^2$ based on published estimates derived from mean annual air temperature (see discussion in Section 2.3.4) .

There are six main limitations of our study. First, thermokarst processes due to melt of excess ground ice (ice wedges or lenses) are not simulated. As maps of ground ice extent improve (e.g. O'Neill et al., 2018) and become more suitable for use as a model geophysical field, parameterizations such as Lee et al. (2014) could be incorporated. Second, our treatment of mosses and their impact is simplistic. A more comprehensive approach such as the LiBry model (Porada et al., 2016) would allow for dynamic moss extents and more bryophyte subtypes including lichens. Third, the plant functional types used here are not specific to the Arctic and do not include shrubs. Shrubs, in particular, are presently expanding and have complex impacts upon Arctic regions (e.g. Figure 3 in Myers-Smith et al., 2011). Fourth, orographic influences on permafrost such as slope and aspect were not resolved. Fifth, inland water bodies and their impact upon ground thermal regimes were not considered. Finally, the influence of sub-grid heterogeneity was ignored as permafrost in the model grids is binary thus excluding the simulation of discontinuous permafrost. With regard to the influence of sub-grid heterogeneity, the standard deviation of ALT on the 1 km$^2$ - 1 ha measurement grids at the GTN-P ALT sites, the spread in MAE in grid cells with multiple GTN-P ALT sites, and the SPSMPS collection of 35 boreholes over a 1200 km$^2$ study area indicate that it is likely difficult to reduce the wMAE of ALT or borehole temperature much further, given the size of the model grid cells (ca. 2.8°). Based on the model physics performance

presented here, it appears that with the modifications described above, the land surface scheme in CLASS-CTEM is well-suited to provide the physical conditions for simulating carbon fluxes in the permafrost domain.

*Code availability.* CLASS-CTEM is available as a tarball from Melton (2019). The following code tags correspond to experiments in this manuscript (see Table 1) with the most strongly impacted subroutines in parentheses: 1) Base model: 'archive/baseModelPermafrostPhysics',
2) deVries thermal cond.: 'archive/soilthermalcond' (TPREP), 3) Tian16 thermal cond.: 'archive/Tian16SoilThermalCond (TPREP), 4) Snow cover: Yang97/Brown03: 'archive/snowcov_changes' (CLASSA), 5) Fresh snow density: 'archive/snowdens' (CLASSI), 6) Snow albedo decay: 'archive/snowalbedorefresh' (CLASSA,SNOALBA), 7) Super-cooled water: 'archive/supercooledH2O' (CLASSB,TMCALC,TWCALC), and 8) Modif. hydrology: 'archive/arman' (GRDRAN,GRDINFL). The model manual is located within the code repository (/documentation/html/index.html)

**Appendix A: Description of alternate parameterizations**

## A1    Moss parameterization of Wu et al. (2016)

The simple moss parameterization used here follows Wu et al. (2016) with the exception that our moss layer is non-photosynthesizing. The physical characteristics of the moss layer includes a pore volume of 0.98 $m^3m^{-3}$, liquid water retention capacity of 0.2 $m^3m^{-3}$, the residual liquid water content after freezing or evaporation of 0.01 $m^3m^{-3}$, the Clapp and Hornberger empirical
"b" parameter set to 2.3, a soil moisture suction at saturation of 0.0103 m, a saturated hydraulic conductivity of $1.83 \times 10^{-3}$ $m\ s^{-1}$, a volumetric heat capacity of $2.5 \times 10^{-6}$ $J\ m^{-3}K^{-1}$, with the thermal conductivity of the moss set to that of organic matter (0.25 $W\ m^{-1}\ K^{-1}$).

## A2    Soil thermal conductivity

CLASS-CTEM calculates the thermal conductivities of organic and mineral soils following Côté and Konrad (2005). The soil
thermal conductivity, $\lambda$ ($W\ m^{-1}\ K^{-1}$), is modelled via a relative thermal conductivity, $\lambda_r$, which varies between a value of 1 at saturation and 0 for dry soils:

$$\lambda = [\lambda_{sat} - \lambda_{dry}]\lambda_r + \lambda_{dry} \tag{A1}$$

Using the following generalized relationship, the relative thermal conductivity is obtained from the degree of saturation (the water content divided by the pore volume), $S_r$ (unitless):

$$\lambda_r = \frac{\kappa S_r}{[1 + (\kappa - 1)S_r]} \tag{A2}$$

Based on the soil characteristics and state, the empirical coefficient, $\kappa$ ($W\ m^{-1}\ K^{-1}$), takes the following values:

1. Unfrozen coarse mineral soils: $\kappa = 4.0$

2. Frozen coarse mineral soils: $\kappa = 1.2$

3. Unfrozen fine mineral soils: $\kappa = 1.9$

4. Frozen fine mineral soils: $\kappa = 0.85$

5. Unfrozen organic soils: $\kappa = 0.6$

6. Frozen organic soils: $\kappa = 0.25$

The dry thermal conductivity, $\lambda_{dry}$, is calculated via an empirical relationship using the pore volume, $\theta_p$ (m$^3$ m$^{-3}$), with different coefficients for organic and mineral soils:

$$\lambda_{dry,mineral} = 0.75 e^{(-2.76\theta_p)} \tag{A3}$$

$$\lambda_{dry,organic} = 0.30 e^{(-2.0\theta_p)} \tag{A4}$$

While the saturated thermal conductivity, $\lambda_{sat}$, is calculated by Côté and Konrad (2005) as a geometric mean of the conductivities of the soil components, other studies (e.g. Zhang et al., 2008) have found the linear averaging used by de Vries (1963) to be generally more accurate and this approach has been adopted by CLASS-CTEM,

$$\lambda_{sat,unfrozen} = \lambda_{liq}\theta_p + \lambda_s(1 - \theta_p) \tag{A5}$$

$$\lambda_{sat,frozen} = \lambda_{ice}\theta_p + \lambda_s(1 - \theta_p) \tag{A6}$$

where $\lambda_{ice}$ is the thermal conductivity of ice, $\lambda_{liq}$ is that of liquid water and $\lambda_s$ is that of the soil solid particles.

Exp. *deVries thermal cond.* replaces the CLASS-CTEM default soil thermal conductivity parameterization with that of de Vries (1963):

$$\lambda = \frac{\lambda_{liq}\theta_{liq} + f_a\lambda_a\theta_a + f_s\lambda_s\theta_s}{\theta_{liq} + f_a\theta_a + f_s\theta_s} \tag{A7}$$

where the $a$ subscript denotes the air component, $\theta$ is the volumetric fraction, and $f$ is the 'weighting' factor (unitless) which is given by:

$$f_s = \frac{1}{3}\left[\frac{2}{1+0.125(\frac{\lambda_s}{\lambda_{liq}}-1)} + \frac{1}{1+0.75(\frac{\lambda_s}{\lambda_{liq}}-1)}\right] \tag{A8}$$

$$f_a = \frac{1}{3}\left[\frac{2}{1+g_a(\frac{\lambda_a}{\lambda_{liq}}-1)} + \frac{1}{1+(1-2g_a)(\frac{\lambda_a}{\lambda_{liq}}-1)}\right] \tag{A9}$$

where $g_a$ represents a unit-less empirical air pore-shape factor,

$$g_a = \begin{cases} 0.333 - (0.333 - 0.035)\frac{\theta_a}{\theta_p}, & \theta_{liq} > 0.09 \\ 0.013 + 0.944\theta_{liq}, & \theta_{liq} \leq 0.09 \end{cases} \tag{A10}$$

An alternate approach is tested in Exp. *Tian16 thermal cond.*. The Tian et al. (2016) thermal conductivity parameterization is based upon the de Vries (1963) formulation, but simplifies and extends it to both frozen and unfrozen soils. In their formulation, Tian et al. adapt equation A7 to include ice and organic matter as,

$$\lambda = \frac{\lambda_{liq}\theta_{liq} + f_{ice}\lambda_{ice}\theta_{ice} + f_a\lambda_a\theta_a + f_s\lambda_s\theta_s + f_{organic}\lambda_{organic}\theta_{organic}}{\theta_{liq} + f_{ice}\theta_{ice} + f_a\theta_a + f_s\theta_s + f_{organic}\theta_{organic}} \tag{A11}$$

for wet soil whereas the thermal conductivity of completely dry soils is calculated by,

$$\lambda = 1.25\frac{f_a\lambda_a\theta_a + f_s\lambda_s\theta_s + f_{organic}\lambda_{organic}\theta_{organic}}{f_a\theta_a + f_s\theta_s + f_{organic}\theta_{organic}} \tag{A12}$$

The Tian et al. formulation also modifies the pore-shape factor (equation A10) to be,

$$g_a = 0.333 - \left(1 - \frac{\theta_a}{\theta_p}\right) \tag{A13}$$

for air and

$$g_{ice} = 0.333 - \left(1 - \frac{\theta_{ice}}{\theta_p}\right) \tag{A14}$$

for ice. Tian et al. (2016) introduce a shape factor for ellipsoidal soil particles, $g_m$ as,

$$g_m = g_{sand}\theta_{sand} + g_{silt}\theta_{silt} + g_{clay}\theta_{clay} \tag{A15}$$

where $g_{sand}$ is 0.182, $g_{silt}$ is 0.00775, and $g_{clay}$ is 0.0534. The shape factor for organic soils, $g_{organic}$, is set to 0.5. The same 'weighting' factor is used for ice, air, organic and mineral soil components and left unchanged from equation A9.

## A3 Snow cover fraction

CLASS-CTEM relates snow depth, ($d_{snow}$; m), to snow cover, ($f_{snow}$; fraction), via a linear function (Supplement Figure 2) (Verseghy, 2017),

$$f_{snow} = min\left[1, \left(\frac{d_{snow}}{d_0}\right)\right] \tag{A16}$$

where $d_0$ is a limiting snow depth assigned a value of 0.1 m. Exp. 'Snow cover:Yang97' changes the CLASS-CTEM linear function to a hyperbolic tangent function (Yang et al., 1997),

$$f_{snow} = tanh\left(\frac{d_{snow}}{d_0}\right) \tag{A17}$$

Another alternative parameterization for snow cover from snow depth was proposed by Brown et al. (2003), which was not evaluated in MacDonald (2015). This relation was developed based on analysis of a global gridded snow water equivalent product designed to evaluate GCMs. Exp 'Snow cover:Brown03' tests the impact of that parameterization by changing the snow cover function to the proposed exponential form (Brown et al., 2003),

$$f_{snow} = 1 - 0.01(15 - 100d_{snow})^{1.7}. \tag{A18}$$

## A4 Fresh snow density

The density of freshly fallen snow is related to its ice-crystal structure and the volume of the ice crystal that is occupied by air. Generally, snow density is the result of 1) processes occurring in the cloud that affect the size and shape of the growing ice crystals, 2) processes that modify the crystal as it falls, and 3) compaction on the ground due to prevailing weather conditions and metamorphism in the snowpack (Roebber et al., 2003).

Fresh snow density ($\varrho$; kg m$^{-3}$) in CLASS-CTEM is calculated based on air temperature ($T_a$; K). For air temperatures below freezing, ($T_f$), a relation from Hedstrom and Pomeroy (1998) is used, while for temperatures at or above freezing CLASS-CTEM uses an equation from Pomeroy and Gray (1995),

$$\varrho = \begin{cases} 67.92 + 51.25e^{\left[\frac{(T_a - T_f)}{2.59}\right]} & T_a < T_f \\ 119.17 + 20(T_a - T_f) & T_a \geq T_f \end{cases} \tag{A19}$$

In Exp. *Fresh snow density*, the effect of wind speed ($u$, m s$^{-1}$) is included following the approach used in the CROCUS model as detailed in Essery et al. (1999) with a minimum density of 50 kg m$^{-3}$ following MacDonald (2015):

$$\varrho = max[50, 109 + 6(T_a - T_f) + 26u^{1/2}] \tag{A20}$$

Wind speed may be considered important in determining fresh snow density as wind speeds greater than approximately 9 m s$^{-1}$ can move ice crystals on the surface leading to crystal fractionation during saltation and surface compaction increasing the snow density (e.g., Gray and Male, 1981, p. 345–350).

## A5   Snow albedo decay

Snow albedo ($\alpha_s$; unitless) decreases as snow ages due to snow grain growth and deposition of soot and dirt (Wiscombe and Warren, 1980). In CLASS-CTEM this process is treated via empirical exponential decay functions (Verseghy, 2017). Freshly fallen snow is given a total albedo ($\alpha_{fs,total}$) value of 0.84, a visible ($\alpha_{fs,visible}$) value of 0.95 and a near-infrared (NIR; $\alpha_{fs,nir}$) value of 0.73 . It is assumed that the same decay function, calculated each timestep ($\Delta t$; 1800 s) applies to all three albedo ranges,

$$\alpha_{s,total}(t+\Delta t) = \alpha_{s,total,old} + [\alpha_{s,total}(t) - \alpha_{s,total,old}]e^{\left(-\frac{0.01\Delta t}{3600}\right)} \tag{A21}$$

If the snowpack temperature is greater than -0.01 °C or the melt rate at the top of the snowpack is not negligible, $\alpha_{s,total,old}$ is set to a value characteristic of melting snow (0.50) otherwise it is set a value representing old, dry snow (0.70). The total albedo at a given time step is converted to those of the visible and NIR ranges for dry snow via,

$$\alpha_{s,visible} = 0.7857\alpha_{s,total} + 0.29 \tag{A22}$$

$$\alpha_{s,nir} = 1.2142\alpha_{s,total} - 0.29 \tag{A23}$$

and for melting snow,

$$\alpha_{s,visible} = 0.9706\alpha_{s,total} + 0.1347 \tag{A24}$$

$$\alpha_{s,nir} = 1.0294\alpha_{s,total} - 0.1347 \tag{A25}$$

Exp. *Snow albedo decay* replaces the CLASS-CTEM exponential decay function with a spectral method based on Wiscombe
and Warren (1980) and adapted for efficiency by Dickinson (1983). This efficient spectral method first calculates the diffuse radiation albedo based on the albedo of fresh snow and the transformed snow age factor ($F_{age}$)

$$\alpha_{dif,visible} = (1 - 0.2F_{age})\alpha_{fs,visible} \tag{A26}$$

$$\alpha_{dif,nir} = (1 - 0.5 F_{age}) \alpha_{fs,nir} \tag{A27}$$

$$F_{age} = \frac{\tau_s}{1 + \tau_s} \tag{A28}$$

where $\tau_s$ is a non-dimensional snow age at each timestep found via

$$\tau_s(t + \Delta t) = \left[ \tau_s(t) + \frac{(r_1 + r_2 + r_3)\Delta t}{\tau_0} \right] \left( 1 - \frac{S_f \Delta t}{\Delta P} \right) \tag{A29}$$

where $r_1$ represents the effects of grain growth due to vapor diffusion as

$$r_1 = e^{\left[ 5000 \left( \frac{1}{T_f} - \frac{1}{T_{g,1}} \right) \right]} \tag{A30}$$

and $r_2 = r_1^{10}$, representing the additional effects at or near the freezing of meltwater on grain growth. $r_3$ represents the effects of soot and dirt and is set to 0.3. $T_{g,1}$ is the temperature of the top soil layer (K), $\tau_0$ is $10^6$ s, $S_f$ is the snowfall rate for that timestep (kg m$^{-2}$ s$^{-1}$), and $\Delta P$ is the snow fall amount threshold (10 kg m$^{-2}$). If, within a timestep, the fresh snowfall

amount exceeds $\Delta P$, the snow age is set to that of new snow ($\tau_s = F_{age} = 0$).

The direct radiation albedos are found by

$$\alpha_{dir,visible} = \alpha_{dif,visible} + 0.4 f(\mu)(1 - \alpha_{dif,visible}) \tag{A31}$$

$$\alpha_{dir,nir} = \alpha_{dif,nir} + 0.4 f(\mu)(1 - \alpha_{dif,nir}) \tag{A32}$$

where $f(\mu)$ is a factor that scales between 0 and 1 to give increased snow albedo due to solar zenith angles exceeding 60 $^\circ$,

calculated as

$$f(\mu) = \max \left[ 0, \frac{1 - 2\cos Z}{1 + b_\mu} \right] \tag{A33}$$

where $Z$ is the solar zenith angle and $b_\mu$ is an adjustable parameter set to 2 following the BATS model (Yang et al., 1997).

## A6    Super-cooled soil water

In experiment *Super-cooled water*, unfrozen soil water in frozen soils is introduced into CLASS-CTEM following Niu and

Yang (2006). Unfrozen water can exist in frozen soils through the capillary and absorptive forces exerted by soil particles on

water in close proximity. The upper limit on the residual amount of water that can remain liquid under given soil temperature and texture conditions is parameterized by Niu and Yang (2006) as,

$$\theta_{liq,max} = \theta_p \left( \frac{-L_f(T_{soil,i} - T_f)}{g\psi_{sat}T_{soil,i}} \right)^{-1/b} \tag{A34}$$

where g is gravitational acceleration (m s$^{-2}$), $L_f$ is the latent heat of fusion (J kg$^{-1}$), and $T_{soil,i}$ is the soil layer temperature (K). According to Romanovsky and Osterkamp (2000) unfrozen water content in moss is negligible so $\theta_{liq,max}$ is set to zero for moss layers.

## A7 Modified hydrology

In Ganji et al. (2015) several changes were implemented in CLASS to address how the model deals with frozen soil water. First, super-cooled soil water was added following Niu and Yang (2006) as described above. Secondly, fractional impermeable area was introduced, also following Niu and Yang (2006), but this has little impact upon our model simulations (discussed in Appendix B). Their final modification was to account for the impact of frozen water on the soil matric potential ($\psi$; m) after Farouki (1981) and Koren et al. (1999) by adding a new term $[(1 + C_k\theta_{ice})^2]$ to the existing CLASS functional relationship,

$$\psi = \psi_{sat} \left( \frac{\theta_{liq}}{\theta_p} \right)^{-b} (1 + C_k\theta_{ice})^2 \tag{A35}$$

where $C_k$ is a constant, set to 8, that accounts for the effect of an increase in specific surface area of soil minerals and liquid water as water freezes and ice forms (Kulik, 1978). $\psi_{sat}$ is the soil matric potential at saturation (m) and $b$ is the Clapp and Hornberger empirical 'b' parameter (unitless) (Clapp and Hornberger, 1978). The calculation of hydraulic conductivity $k; ms^{-1}$ is also modified by multiplication with a similar term $[(1 + C_k\theta_{ice})^{-4}]$,

$$k = k_{sat} \left( \frac{\theta_{liq}}{\theta_p} \right)^{2b+3} (1 + C_k\theta_{ice})^{-4} \tag{A36}$$

where $k_{sat}$ is saturated hydraulic conductivity. The effect of these changes is to generally increase soil matric potential and decrease hydraulic conductivity when ice is present in the soil. These modifications are tested in Exp. *Modif. hydrology*.

## Appendix B: Fractional permeable areas in frozen soils

CLASS-CTEM accounts for the impact of frozen soil water through an empirical correction factor ($f_{ice}$; unitless), according to Zhao and Gray (1997).

$$f_{ice} = \left[ 1 - min \left( 1, \frac{\theta_{ice}}{\theta_p} \right) \right]^2 \tag{B1}$$

This factor is used to correct the calculated soil hydraulic conductivity, $k$ (m s$^{-1}$) which is found via the Clapp and Hornberger (1978) equation:

$$k = f_{ice} k_{sat} \left( \frac{\theta_{liq}}{\theta_p} \right)^{2b+3} \tag{B2}$$

where $k_{sat}$ is the hydraulic conductivity at saturation and $b$ is an empirical parameter. Soil moisture is related to soil matric potential ($\psi$; m) in CLASS-CTEM following Clapp and Hornberger (1978),

$$\psi = \psi_{sat} \left( \frac{\theta_{liq}}{\theta_p} \right)^{-b} \tag{B3}$$

where $\psi_{sat}$ is the saturated soil matric potential (m).

Niu and Yang (2006) parameterize fractional permeable areas in frozen soils. Following their formulation, within a grid cell the permeable ($perm$) and impermeable ($imp$) patches affect the flux of water ($q$; m s$^{-1}$) as

$$q = F_{imp} q_{imp} + (1 - F_{imp}) q_{perm} \tag{B4}$$

where the impermeable grid cell fraction, $F_{imp}$ can be estimated as

$$F_{imp} = e^{-\alpha \left( 1 - \frac{\theta_{ice}}{\theta_p} \right)} - e^{-\alpha} \tag{B5}$$

and $\alpha$ is set to 3 following Niu and Yang (2006). Assuming $q_{imp}$ is set to zero, Niu and Yang parameterize the influence of the permeable areas on hydraulic conductivity can be parameterized as

$$k = (1 - F_{imp}) k_{sat} \left( \frac{\theta_{liq} + \theta_{ice}}{\theta_p} \right)^{2b+3} \tag{B6}$$

while the soil matric potential is calculated as,

$$\psi = \psi_{sat} \left( \frac{\theta_{liq} + \theta_{ice}}{\theta_p} \right)^{-b} \tag{B7}$$

This formulation results in a soil matric potential that is insensitive to ice content within the soil (Supplement Figure 9) which seems unreasonable (see for example Wen et al., 2012). This fact is indeed noted by Ganji et al. (2015) who state that the soil matric potential as defined by Niu and Yang (2006) is not appropriate for the case of frozen soil. The inclusion of $\theta_{ice}$ in the numerator could be a typographical error. If it is removed the hydraulic conductivity and soil matric potential behave quite similarly to the original CLASS relations which make use of the factor $f_{ice}$ in place of $1 - F_{imp}$ (Supplement Figure 10). Testing shows the model is relatively insensitive to the small changes visible in the plots (not shown).

**Table A1.** Ground layer depths and thicknesses for the 20 ground layer configuration.

| Layer number | Thickness (m) | Depth (m) |
| --- | --- | --- |
| 1 | 0.1 | 0.1 |
| 2 | 0.1 | 0.2 |
| 3 | 0.1 | 0.3 |
| 4 | 0.1 | 0.4 |
| 5 | 0.1 | 0.5 |
| 6 | 0.1 | 0.6 |
| 7 | 0.1 | 0.7 |
| 8 | 0.1 | 0.8 |
| 9 | 0.1 | 0.9 |
| 10 | 0.1 | 1.0 |
| 11 | 0.2 | 1.2 |
| 12 | 0.3 | 1.5 |
| 13 | 0.4 | 1.9 |
| 14 | 0.5 | 2.4 |
| 15 | 1.0 | 3.4 |
| 16 | 3.0 | 6.4 |
| 17 | 5.0 | 11.4 |
| 18 | 15.0 | 26.4 |
| 19 | 30.0 | 56.4 |
| 20 | 5.0 | 61.4 |

*Author contributions.* JRM initiated the study, performed the model simulations and analysis and wrote the paper. DV led the development of the CLASS model, conducted initial research into the recommendations of MacDonald (2015) and was liaison to the Sushama group for the work of Arman Ganji. RS-A performed the statistical analysis and plotting for SWE and MODIS albedo. SG provided the Lac de Gras data and participated in discussions around model evaluation. All authors contributed to the final version of the paper.

5    *Competing interests.* None

*Acknowledgements.* We thank the Global Terrestrial Network for Permafrost for generously sharing their data and for making it easily accessible on-line. We thank Dr. Vivek Arora for processing the CRUJRA55 meteorological data, Mr. Ed Chan for processing the MODIS data, and Drs. Christian Seiler and Paul Bartlett for providing comments on a pre-submission version of our MS. Sampling at the Khanovey site is supported by the RuNoCORE CPRU-2017/10015 https://www.siu.no/eng/content/view/full/81242 ; the SAMCoT WP6
10   https://www.ntnu.edu/web/samcot/home and Lomonosov Moscow state university, Geology faculty, permafrost department.

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
