# Peer review of "Improving permafrost physics in the coupled Canadian Land Surface Scheme (v. 3.6.2) and Canadian Terrestrial Ecosystem Model (v. 2.1) (CLASS-CTEM)"

_Geoscientific Model Development, 2019_

## Referee Comment (RC1) · Anonymous Referee #1 · 16 May 2019

Comments:

General

This paper describes improvements and tests of the CLASS-CTEM model under 18 different model configurations and input data sources. The results are compared with site observations of active-layer thickness (ALT) and ground temperatures, and estimates of permafrost extent and snow water equivalent (SWE). Such systematic tests demonstrate the effects of different configurations and data sources on the model behaviour,

therefore, it provides basis for its improvements. Such information is also useful for the improvement of other land surface models in simulating permafrost regions. The topic of this paper is a good fit of the scope of the journal, and it is worthy to be published. Following are some suggestions, which may be useful to the improvement of its readability.

Major points

1. As the paper has indicated, the major issue is the sub-grid heterogeneity. The site conditions of the observations can be very different from the grid average used in the model. The input climate data for the model are also different from the climate at the observation sites. Thus, the nature of the modelled grid cells and the observation sites are quite different things. However, these multiple site observations are the data currently available, and could somewhat represent the conditions of the grid cell. Therefore, the approach used in this study is useful to check and improve the overall behavior of the model across the permafrost domain. The model is better constrained overall but not necessarily validated. Sections 3.12 and 3.13 indicated this issue. I think this is a limitation or shortcoming of the methodology rather than an excuse of the modelled bias (e.g., the authors seem to argue that the model can represent the reality better (Line 25, page 27)). Such limitation should be indicated clearly and to frame the assumptions/methods better.

2. The results show that the simulated ALT is improved but no significant improvement in ground temperatures according to Figure 5. Since ground temperature is the principal state variable of the model, it would be useful to provide more information about it. Figure 4 is a nice demonstration of the improvement of the modelled ALT. Similar figures should be provided for the ground temperatures about their bias distributions. If they take too much space, you can put some of them in the Appendix.

3. The paper is too long and some analysis is not very concise. Some parts of the analysis can be reduced, such as latent heat, sensible heat, and albedo as there are

no observations for these variables anyway. Even the discharge shown in Figure 9 is not necessary. It is better just focusing on ALT, ground temperature and permafrost extent.

Minor points

1. P.1, Lines16-18: "with seasonal . . .at the sites". Not clear.

2. P.1 Lines 18-19: "Sub-grid heterogeneity estimates were derived from the standard deviation of ALT on the 1 km2 measurement grids at the GTN-P ALT sites". Its sounds like you estimate the sub-grid heterogeneity for all the regions. That is not the case.

3. P2. Line 12-13. "Since the carbon stored in frozen soils is only accessible to microbial respiration once soils thaw". Soil respiration has been observed when soil is frozen although it is low. The word "only" is too restrictive.

4. P3, Line 5: "that four be considered", Four types parameterizations?

5. P.4, Line 33: "configuration. (e.g", delete the '.'

6. Page 5: "The first seven experiments" in line 5 is too far away from "the second series of experiments" in line 32. It is better to put them closer.

7. Page 7, Line 33: It is only for ground temperature which is converted to monthly averages? ALT is the annual maximum thaw depth. "The closest CLASS-CTEM grid cell to the ALT site's location" Why it is closest to a grid cell not within a grid cell?

8. Page 9, Line 24: delete the repeating "are".

9. Page 10, Lines 11-12: "but comparing . . .. Poor agreement", not very clear.

10. Page 14, Fig. 3: The Y-axis is called 'residual'. Is it the difference between simulated and observed ALT as indicated in the text? If so, it would be clearer to indicate that. The X-axis is 'Ground depth (m)'. I think it is active-layer thickness although not sure it is modelled or observed. If so, it is better to say that.

11. Page 15, Lines 6-11 (even to line 18). It is better to put this paragraph to the section 2.2 (study design).

12. Page 6, Figure 6. The top line "Depth = 0.05 – 5m |Season = JDF . . .", 5m should be 0.5m.

13. Page 21, Figure 7. The first sentence of the caption is not clear.

14. Page 27, Line 25, The word 'best' is no appropriate.

---

## Short Comment (SC1) · 24 May 2019

I am writing as an executive editor of GMD to highlight an issue with the code availability section which needs to be remedied in the revised manuscript.

I appreciate the lengths to which you have gone to identify the exact versions of the code corresponding to the experiments in the manuscript. There remain two concerns.

[Figure]

**Code only on GitLab**

The code reference is to a GitLab repository (in a personal account at that), and the individual tags are git tags. This doesn't provide the persistence required of the data in a journal paper. If the project moves to a different revision control system, or even just to GitHub then the links will go dead. For this reason, the code associated with each of your tags needs to be persistently archived. Many authors find Zenodo (https: //zenodo.org) a good solution for this, it basically comes down to uploading a tarball and metadata (or you could push a copy of your repo to GitHub and use the automated GitHub-Zenodo integration). Note that you are still positively encouraged to provide a link to the preferred download location for your software (which might well be GitLab), but this should be in addition to citing persistent archives.

**Insufficient documentation to reproduce**

When I click through to your GitLab page, I can't find documentation which would tell me how to build the model and run the experiments in the paper. The readme file claims to contain links to documentation, but these are broken. A link to a wiki is also not very persistent: would a user coming to this after the project has ended be able to work out how to re-run the experiments in this manuscript?

Please ensure that both your code and documentation are preserved in persistent, public archives.

---

## Referee Comment (RC2) · Anonymous Referee #2 · 29 May 2019

The manuscript presents an improved model version of the Canadian CLASS-CTEM model with respect to permafrost physics. The authors have done a great and extensive job dealing with the uncertainties of heat transfer within cold soils. Several tests were performed to see the optimised results and compared to observational datasets. The improved model version is a valuable formulation to be used in offline and coupled simulations. The analysis in the manuscript can also help identify other modelling groups for better physical formulations. The topic and the presentation fits the journal's scope, yet I have some minor suggestions to the authors to make the paper bit more

easy to read through:

1. The extent of statistical analysis is way too long in the manuscript. I strongly suggest to move some of them to supplementary materials to make the actual paper more on point and show the most optimal formulations inside the main manuscript.

2. I agree with the authors to focus on the big scale improvements rather than grid point based comparisons but to actually identify the process improvements, it would be useful to show two or three selected grid points and compare the surface (∼10-20cm) soil temperature time series for different experiments in addition to the borehole temperature comparisons in fig11.

3. To better quantify the snow pack improvement process, it would also help to show comparisons of snow depth with the observed values (if it exists). Since snow insulation plays a major role in freeze/thaw periods, the simulated snow depth should be investigated.

―――――――――――――――――――――――――

---

## Referee Comment (RC3) · Anonymous Referee #3 · 30 May 2019

In this paper the authors test and evaluate a wide range of improvements to permafrost physics in the CLASS-CTEM land surface model (which is part of the Canadian Earth System Model, CanESM). While there is nothing especially ground-breaking, this is a comprehensive and thorough assessment bringing together many different, disparate developments into a single framework and I believe is worthy of publication.

CLASS-CTEM consists of two components: broadly, CLASS does the physical calculations and CTEM performs the carbon cycle calculations. The authors improve the simulation of permafrost physics in CLASS-CTEM with a series of model developments

which are successively evaluated (against multiple observations) and discussed. The default soil scheme has only three vertical layers which leads to a poor simulation of permafrost dynamics. Therefore, very reasonably, the first improvement extends the soil column and adds more layers. A moss layer is added to the surface, as this has previously been shown to improve simulation of soil temperature and freeze/thaw dynamics. Furthermore, the authors experiment with the depth to which the water can penetrate in the soil, and the impact of different driving data sets. Having established a baseline simulation based on what they consider to be these essential improvements, the authors then test a number of further developments in the representation of snow, hydrology and heat transfer. This includes, for example, allowing the presence of liquid water below zero degrees celsius ('supercooled' water), as in real life soils. Of the developments tested, this is considered to make the greatest improvement and therefore to be incorporated into the standard model version. The final simulation of permafrost by CLASS-CTEM is an improvement on the initial simulation, although the capacity for evaluation is somewhat limited by the disparity between the extremely large grid cell size in comparison to site-level observations.

While this paper is relatively clear and well-written, I believe that it can and should be significantly improved prior to publication.

GENERAL COMMENTS

- In terms of experiments included, since de Vries thermal conductivity is considered to be physically unrealistic, I don't see the need to include this experiment.

- The paper is a bit too long and the clarity could be improved. The statistics don't need to be written out in so much detail in the text. As an example, the section 3.11 for 'modified hydrology' could be reduced to something along the lines of: "This development generally reduces water mobility (eqs. A35/A36), resulting in wetter soils, which in turn leads to a significantly deeper ALT but minimal impact on soil temperature errors (see Figures 2,4 and 5). Overall it increases the spring runoff and reduces winter

runoff, which degrades model performance compared to runoff observations (Figure 9). Therefore, this did not provide any significant improvements." [of course this could be better written, just a suggestion for what info to include -ie numbers not necessary in text, and only including the most crucial points] I would then also suggest adding a comment on the process and why it doesn't work, for example my initial thought might be that while it may be a more realistic parameterisation, the fact that the model does not account for macropore flow / large scale cracks and defects in the ground means that the default parameterisation may be providing a compensation for this.

In general I would like to see more reference to the processes and to the direction of the biases, rather than to absolute error, as this gives clearer information about the processes and why each development is having a particular effect. I might suggest to include a directional error in the error tables (bias in annual mean?) as well as just MAE.

- I don't think detailing the heat flux and LAI for each experiment is helpful to the main message. You could summarize heat fluxes and LAI as a separate section, for example noting that in warmer, wetter simulations, the LAI is generally larger.

- In Section 3.13 I suggest adding a further comment on the potential ALT biases. I have noticed from using ALT data that the maximum that depth in the datasets is sometimes not the actual end of season maximum because the field campaigns have not continued right until the end of the season, so this could be the cause of apparently discrepancy between soil temperatures and ALT's.

- It appears that most of the developments are detailed in the appendix, but it doesn't appear to include details of the moss parameterisation. Please add something about the changes to thermal/hydraulic parameters for reproducibility (apologies if I missed this).

- I would expect 'mean absolute error' to be higher for the air than for the soil, because variations in air temperature are typically much larger (both seasonally and on short

timescales). Therefore I don't completely agree with the argument on page 27 lines 3-9.

FIGURES

I find Figure 3 quite confusing and not entirely helpful. It is not clear what either of the axes are: The residual I guess is obs-model, but it's not specified in the text, and it's not at all clear what the x-axis is showing. Is this the observed ALT? If so I think it would be better, and show the same information, if you just to plot model ALT against observed ALT. I am also not entirely convinced by all the discussions that relate to that figure in terms of 'biases with depth', since there are altogether not many points on the plots and the trends do not appear to be very strong. Perhaps these discussions could be removed or modified to focus on the more significant impacts of the developments. For example, the 20 layer simulation has in general too deep ALT- would be enough information.

Figure 7 doesn't add much. We can see basically the same thing but more clearly on Figure 4. If it is to be included, I would suggest in the appendix or supplementary.

Figure 11: I am confused as to why the simulation with moss is warmer than the simulation with no moss. Are these labelled correctly?

MINOR COMMENTS

" with seasonal wMAE values for the shallow surface layers of the revised model simulation at most 1.2 âŮęC greater than those calculated for the model driving screen-level air temperature compared to observations at the sites" - This sentence is difficult to parse. Perhaps by removing the last part ("compared to observations at the sites"), it would make more sense.

Page 7 line 14: How is the 1851-1900 part different from just doing two more spinups? Is the CO2 varying? CO2 forcing dataset is not mentioned but should be included - please add.

[Figure]

Page 8 line 10: "filtered out"- not sure what that means? Does it mean removing the same values from model as are missing in observations?

Please include how you define permafrost presence in a grid cell (again, apologies if I missed this).

Page 9 line 31. 'more grid cells' -> 'additional grid cells' (for clarity)

Page 13 line 7-8. I would expect the absence of water and latent heat to make active layer too deep, because latent heat suppresses the seasonal cycle. But since latent heat works equally in both directions (applicable both in freezing and thawing periods), I am not convinced that its absence would lead to warmer soils. Indeed I am not seeing a major bias in soilgrids vs 20 layers on Figure 6. Perhaps reconsider this sentence.

P 13 line 25: "accurately" -> "accurately simulating"

P 15 line 17-18. Soilgrids has extremely deep soils in West Siberia/Urals: Can you find some reference or ask someone who's been there whether this is at all realistic? Having claimed this is a better validated dataset it would be helpful to provide evidence of this.

P17 line 15-16. Cold bias due to too much moss is a reasonable assumption, but coupled with the fact that there doesn't seem to be a major warm bias without it (fig 6), and that you would overall expect observations to be biased cold (as written in 3.13), I am not totally convinced here. More consideration of processes and what is missing may be helpful.

P29 line 21-22 "thus excluding the simulation of taliks". I am confused by this statement. A talik could be simulated in a 1D model. Discontinuous permafrost, on the other hand, couldn't be. Please could you either clarify or rephrase.

Hope you find these comments helpful to improve an already good manuscript.

---

## Author Comment (AC1) · 15 Aug 2019

**Reviewer reply for Permafrost Physics**

Dear Dave Lawrence,

We wish to thank our three reviewers and Executive Editor David Ham for their time and considered comments on our manuscript. Below we lay out their comments in full and include our replies in bold font. In response to reviewer comments the revised manuscript has been significantly trimmed down. Some material, which was less essential to the main discussion, has been moved to supplementary material, one figure has been removed and another moved to the appendix. We believe the paper now presents the main points in a more concise manner while still retaining the relevant details.

**Executive Editor David Ham**

I am writing as an executive editor of GMD to highlight an issue with the code availability section which needs to be remedied in the revised manuscript. I appreciate the lengths to which you have gone to identify the exact versions of the code corresponding to the experiments in the manuscript. There remain two concerns.

**Code only on GitLab**

The code reference is to a GitLab repository (in a personal account at that), and the individual tags are git tags. This doesn't provide the persistence required of the data in a journal paper. If the project moves to a different revision control system, or even
just to GitHub then the links will go dead. For this reason, the code associated with each of your tags needs to be persistently archived. Many authors find Zenodo ([https://zenodo.org](https://zenodo.org)) a good solution for this, it basically comes down to uploading a tarball
and metadata (or you could push a copy of your repo to GitHub and use the automated GitHub-Zenodo integration). Note that you are still positively encouraged to provide a link to the preferred download location for your software (which might well be GitLab),
but this should be in addition to citing persistent archives.

**We have deposited our code on Zenodo. The link is provided in the revised MS.**

**Insufficient documentation to reproduce**

When I click through to your GitLab page, I can't find documentation which would tell me how to build the model and run the experiments in the paper. The readme file claims to contain links to documentation, but these are broken. A link to a wiki is also not very persistent: would a user coming to this after the project has ended be able to work out how to re-run the experiments in this manuscript? Please ensure that both your code and documentation are preserved in persistent, public archives.

**The manual is within the code repository, not the gitlab wiki. Within the code repository it is /documentation/html/index.html. Since we structured the code and documentation this way, the manual is always part of the codebase. Our documentation is then also part of the Zenodo repository. We have made this clear in the revised manuscript.**

**Anonymous Referee #1**

**General**

This paper describes improvements and tests of the CLASS-CTEM model under 18 different model configurations and input data sources. The results are compared with site observations of active-layer thickness (ALT) and ground temperatures, and estimates of permafrost extent and snow water equivalent (SWE). Such systematic tests demonstrate the effects of different configurations and data sources on the model behaviour,therefore, it provides basis for its improvements. Such information is also useful for the improvement of other land surface models in simulating permafrost regions. The topic of this paper is a good fit of the scope of the journal, and it is worthy to be published. Following are some suggestions, which may be useful to the improvement of its readability.

**Major points**

1. As the paper has indicated, the major issue is the sub-grid heterogeneity. The site conditions of the observations can be very different from the grid average used in the model. The input climate data for the model are also different from the climate at the observation sites. Thus, the nature of the modelled grid cells and the observation sites are quite different things. However, these multiple site observations are the data currently available, and could somewhat represent the conditions of the grid cell. Therefore, the approach used in this study is useful to check and improve the overall behavior of the model across the permafrost domain. The model is better constrained overall but not necessarily validated. Sections 3.12 and 3.13 indicated this issue. I think this is a limitation or shortcoming of the methodology rather than an excuse of the modelled bias (e.g., the authors seem to argue that the model can represent the reality better (Line 25, page 27)). Such limitation should be indicated clearly and to frame the assumptions/methods better.

   **Determining how well a model performs for the physics of simulating permafrost processes is challenging. Due to the slow response times of permafrost thaw at depth there presently exist no long term observations of both meteorology (for model forcing) and soil temperatures (for evaluation) that are of suitable length to allow a proper test of the model performance. As a result it is difficult to get around the two linked problems raised by the reviewer: sub-grid heterogeneity in both the model drivers and the site-level conditions that we are hoping to simulate. We have attempted to demonstrate how sub-grid heterogeneity could impact our ability to evaluate our model (Section 3.12). Based on our analysis at the GTN-P borehole sites and the SPSMPS borehole cluster, we find that the influence of sub-grid heterogeneity makes it difficult to further decrease model bias with the observational data available. An alternative methodology could be to run the model at a single site and do an in-depth analysis of the model performance against the observations being careful to set the model up to the conditions at the site (soil textures, vegetation, permeable depths, etc.). However, as we state in our Introduction (p 3 l 25), model performance at a single site is not necessarily indicative of performance over large regions. So while we agree that our methodology does not allow a perfect validation of the model's performance, we believe it allows the most robust estimate of the model performance with the available observational information. Note also that we are not attempting to use our attempt to quantify sub-grid heterogeneity effects as an excuse for any model bias but rather we are attempting to contextualize the bias and bring that through to the interpretation of our results.**

   **To clarify, we don't state that our model can represent the reality better, but rather we state:**

**"In summary, it is likely that a slightly positive model bias, i.e. higher temperatures and greater ALT simulated than observed, would correspond to a model that best represents reality."**
**This statement suggests that due to the factors causing sampling bias in the ALT and borehole observations (as outlined in Section 3.13), a perfect model would simulate slightly higher temperature and greater ALT than the observations.**

2. The results show that the simulated ALT is improved but no significant improvement in ground temperatures according to Figure 5. Since ground temperature is the principal state variable of the model, it would be useful to provide more information about it. Figure 4 is a nice demonstration of the improvement of the modelled ALT. Similar figures should be provided for the ground temperatures about their bias distributions. If they take too much space, you can put some of them in the Appendix.
**Actually Figure 5 demonstrates that across all seasons and all depths the model simulated soil temperatures for the SoilGrids simulation are improved compared to the Base model (SoilGrids+Moss is only at certain depths/seasons). There are 105 106 borehole observations from 132 GTN-P sites used in this evaluation. An additional look at the ground temperatures is provided by Figure 6 with Gaussian KDE for the GTN-P sites. A Gaussian KDE is a similar demonstration of bias distribution to that presented in Figure 4 however the KDE plots are made up of many more data points as there are more borehole observations available than ALT. For example, the 0.05 - 0.5 m depth KDE contains 47 079 data points. Because the KDE becomes too difficult to parse with too many experiments plotted we have restricted it to only a few experiments being shown.**

3. The paper is too long and some analysis is not very concise. Some parts of the analysis can be reduced, such as latent heat, sensible heat, and albedo as there are no observations for these variables anyway. Even the discharge shown in Figure 9 is not necessary. It is better just focusing on ALT, ground temperature and permafrostextent.
**We agree and have streamlined the revised MS. The variables without observations for comparison have been moved to a supplement. We have retained the discharge as there are observations for comparison and several of the changes influence hydrology so this is a means to look at their impact.**

**Minor points**

1. P.1, Lines16-18: "with seasonal . . .at the sites". Not clear.
**We have attempted to improve the clarity by rewording from:** *"with seasonal wMAE values for the shallow surface layers of the revised model simulation at most 1.2 ($^{\circ}$)C greater than those calculated for the model driving screen-level air temperature compared to observations at the sites"*, **to:** *"with seasonal wMAE values for the shallow surface layers of the revised model simulation at most 3.7 ($^{\circ}$)C, which is 1.2 ($^{\circ}$)C more than the wMAE of the screen-level air temperature used to drive the model as compared to site-level observations (2.5 ($^{\circ}$)C"*

2. P.1 Lines 18-19: "Sub-grid heterogeneity estimates were derived from the standard deviation of ALT on the 1 km2 measurement grids at the GTN-P ALT sites". Its sounds like you estimate the sub-grid heterogeneity for all the regions. That is not the case.

Unfortunately, we don't quite understand this comment. That sentence reads in full: "*Sub-grid heterogeneity estimates were derived from the standard deviation of ALT on the 1 km(^2) - 1 ha measurement grids at the GTN-P ALT sites, the spread in wMAE in grid cells with multiple GTN-P ALT sites, as well as from 35 boreholes measured within a 1200 km(^2) region as part of the Slave Province Surficial Materials and Permafrost Study.*". The sentence thus explicitly states where the sub-grid heterogeneity estimates were derived from. Note that now we clarify that we used both 1 ha and 1 km(^2) sampling sites.**

3. P2. Line 12-13. "Since the carbon stored in frozen soils is only accessible to microbial respiration once soils thaw". Soil respiration has been observed when soil is frozen although it is low. The word "only" is too restrictive.
   **Indeed, reworded to: "*Since the carbon stored in frozen soils becomes readily accessible to microbial respiration once soils thaw*"**

4. P3, Line 5: "that four be considered", Four types parameterizations?
   **Yes, parameterizations. Original sentence: "He investigated 15 alternative parameterizations relating to the model physics and concluded by recommending that four be considered", for better clarity rephrased to: "He investigated 15 alternative parameterizations relating to the model physics and concluded by recommending that four of those be considered"**

5. P.4, Line 33: "configuration. (e.g", delete the '.'
   **Thanks, removed the comma**

6. Page 5: "The first seven experiments" in line 5 is too far away from "the second series of experiments" in line 32. It is better to put them closer.
   **Good suggestion. We have added to line 32: *"Whereas the first series of experiments just described investigated aspects of the model setup, the second series of experiments investigates alternative parameterizations and uses the 'SoilGrids+Moss' experiment as a starting point"***

7. Page 7, Line 33: It is only for ground temperature which is converted to monthly averages? ALT is the annual maximum thaw depth. "The closest CLASS-CTEM grid cell to the ALT site's location" Why it is closest to a grid cell not within a grid cell?
   **ALT is the active layer thickness. This does not imply that it is the annual maximum. Many of the sampling campaigns for ALT had one or two samples per year. It is possible they were unable to time the annual maximum of ALT in their sampling thus we compared the model to observations on a monthly basis to be most comparable. Regarding the closest/within question, we have changed the sentence for clarity to: "The closest grid cell was determined from the centre of the model grid cells to the ALT sampling location and ..."**

8. Page 9, Line 24: delete the repeating "are".
   **Thanks, done.**

9. Page 10, Lines 11-12: "but comparing . . .. Poor agreement", not very clear.
   **Reworded to: "*Owing to the coarseness of the model grid CLASS-CTEM is not able to simulate isolated or sporadic permafrost. For regions of discontinuous and continuous permafrost, comparing the estimated distribution of \citet{Brown1997-un} to the modelled ALT indicates poor agreement.*"**

10. Page 14, Fig. 3: The Y-axis is called 'residual'. Is it the difference between simulated and observed ALT as indicated in the text? If so, it would be clearer to indicate that. The X-axis is 'Ground depth (m)'. I think it is active-layer thickness although not sure it is modelled or observed. If so, it is better to say that.

    **Another reviewer found this figure confusing. Since it was not fundamental to our paper, we have removed it.**

11. Page 15, Lines 6-11 (even to line 18). It is better to put this paragraph to the section 2.2 (study design).

    **We prefer our original placement as it provides a literature overview of previous work that has demonstrated the importance of increasing the ground column depth and number of ground layers.**

12. Page 6, Figure 6. The top line "Depth = 0.05 – 5m |Season = JDF . . .", 5m should be 0.5m.

    **Yes, thank you. Corrected.**

13. Page 21, Figure 7. The first sentence of the caption is not clear.

    **We have attempted to improve clarity by rewording from: "*ALT differences (meters) for experiments that are based on the model setup of 'SoilGrids+Moss' (see Table \ref{explist}) compared to the 'SoilGrids+Moss' simulation.*" to: "*Differences (meters) between the 'SoilGrids+Moss' simulated ALT and the ALT simulated by the alternative parameterization experiments (based on the model setup of 'SoilGrids+Moss', see Table \ref{explist})*"**

14. Page 27, Line 25, The word 'best' is no appropriate.

    **See our reply to Major Point #1.**

**Anonymous Referee #2**

The manuscript presents an improved model version of the Canadian CLASS-CTEM model with respect to permafrost physics. The authors have done a great and extensive job dealing with the uncertainties of heat transfer within cold soils. Several tests were performed to see the optimised results and compared to observational datasets. The improved model version is a valuable formulation to be used in offline and coupled simulations. The analysis in the manuscript can also help identify other modelling groups for better physical formulations. The topic and the presentation fits the journal's
scope, yet I have some minor suggestions to the authors to make the paper bit more easy to read through:

1. The extent of statistical analysis is way too long in the manuscript. I strongly suggest to move some of them to supplementary materials to make the actual paper more on point and show the most optimal formulations inside the main manuscript.

    **The revised MS has been streamlined. We have moved the De Vries experiment to the supplement, moved the variables without an observational constraint to the supplement, and made the text more concise.**

2. I agree with the authors to focus on the big scale improvements rather than grid point based comparisons but to actually identify the process improvements, it would be useful to show two or three selected grid points and compare the surface (~10 - 20cm) soil temperature time series for

different experiments in addition to the borehole temperature comparisons in fig 11.

**While we are glad that the reviewer appreciates our approach to focus on large scale improvements and comparisons across many different locations, we don't feel that showing surface changes at a few selected grid points would offer much additional insights. Additionally the paper, as mentioned by all reviewers, is already too long. Adding a new figure showing changes at shallow depths, with accompanying text for analysis, would add significant length to an already long manuscript.**

3. To better quantify the snow pack improvement process, it would also help to show comparisons of snow depth with the observed values (if it exists). Since snow insulation plays a major role in freeze/thaw periods, the simulated snow depth should be investigated.

**We agree that snow pack changes are important. The snow pack improvements in our manuscript have been suggested by earlier detailed studies and here, only their effect on permafrost is evaluated. This is why we have used what we believe to be the best available snow product (Blended5-SWE) to evaluate modelled changes in SWE. We are not aware of any large-scale snow depth products. However, snow depth and SWE are linked so looking at SWE from the model versus observation-based datasets is still valuable.**

**Anonymous Referee #3**

In this paper the authors test and evaluate a wide range of improvements to permafrost physics in the CLASS-CTEM land surface model (which is part of the Canadian Earth System Model, CanESM). While there is nothing especially ground-breaking, this is a comprehensive and thorough assessment bringing together many different, disparate developments into a single framework and I believe is worthy of publication.

CLASS-CTEM consists of two components: broadly, CLASS does the physical calculations and CTEM performs the carbon cycle calculations. The authors improve the simulation of permafrost physics in CLASS-CTEM with a series of model developments which are successively evaluated (against multiple observations) and discussed. The default soil scheme has only three vertical layers which leads to a poor simulation of permafrost dynamics. Therefore, very reasonably, the first improvement extends the soil column and adds more layers. A moss layer is added to the surface, as this has previously been shown to improve simulation of soil temperature and freeze/thaw dynamics. Furthermore, the authors experiment with the depth to which the water can penetrate in the soil, and the impact of different driving data sets. Having established a baseline simulation based on what they consider to be these essential improvements, the authors then test a number of further developments in the representation of snow, hydrology and heat transfer. This includes, for example, allowing the presence of liquid water below zero degrees celsius ('supercooled' water), as in real life soils. Of the developments tested, this is considered to make the greatest improvement and therefore to be incorporated into the standard model version. The final simulation of permafrost by CLASS-CTEM is an improvement on the initial simulation, although the capacity for evaluation is somewhat limited by the disparity between the extremely large grid cell size in comparison to site-level observations. While this paper is relatively clear and well-written, I believe that it can and should be significantly improved prior to publication.

**GENERAL COMMENTS**

- In terms of experiments included, since de Vries thermal conductivity is considered to be physically unrealistic, I don't see the need to include this experiment.

  **We have moved most of this to the supplement. While we agree it is not a viable parameterization we wish to include its evaluation as it was recommended for incorporation in CLASS-CTEM by a previous study so we want to demonstrate it is not a suitable addition.**

- The paper is a bit too long and the clarity could be improved. The statistics don't need to be written out in so much detail in the text. As an example, the section 3.11 for 'modified hydrology' could be reduced to something along the lines of: "This development generally reduces water mobility (eqs. A35/A36), resulting in wetter soils, which in turn leads to a significantly deeper ALT but minimal impact on soil temperature errors (see Figures 2,4 and 5). Overall it increases the spring runoff and reduces winter runoff, which degrades model performance compared to runoff observations (Figure 9). Therefore, this did not provide any significant improvements." [of course this could be better written, just a suggestion for what info to include -ie numbers not necessary in text, and only including the most crucial points] I would then also suggest adding a comment on the process and why it doesn't work, for example my initial thought might be that while it may be a more realistic parameterisation, the fact that the model does not account for macropore flow / large scale cracks and defects in the ground means

  that the default parameterisation may be providing a compensation for this. In general I would like to see more reference to the processes and to the direction of the biases, rather than to absolute error, as this gives clearer information about the processes and why each development is having a particular effect. I might suggest to include a directional error in the error tables (bias in annual mean?) as well as just MAE.

  **In response to the comments are length and clarity, the revised MS has been streamlined with material that is informative, but not necessary to convey the main messages moved to supplementary material. Concerning further commentary on why certain processes were successful or not in improving model performance, this is difficult to convey convincingly. To illustrate, using the same example of the modified hydrology experiment, we are not able to definitively state whether the proposed modifications are indeed more realistic based on our results presented here. While we may have a general opinion that the parameterization may be more realistic and the model could have a compensating error, we are not able to provide proof within the context of this paper. As a result it is difficult to include commentary such as that while lacking a means or simulations to definitively justify our statements. We investigated the idea of a directional error in the error tables however we found the KDE plots to give a much clearer picture of the full spectrum of model bias, i.e. calculating a directional bias tended to give a muddy picture due to cancelling positive and negative biases whereas KDE could show the spectrum of biases without that issue.**

- I don't think detailing the heat flux and LAI for each experiment is helpful to the main message. You could summarize heat fluxes and LAI as a separate section, for example noting that in warmer, wetter simulations, the LAI is generally larger.

  **These variables have been moved to supplementary material to streamline the manuscript.**

- In Section 3.13 I suggest adding a further comment on the potential ALT biases. I have noticed from using ALT data that the maximum that depth in the datasets is sometimes not the actual end of

season maximum because the field campaigns have not continued right until the end of the season, so this could be the cause of apparently discrepancy between soil temperatures and ALT's.

**This is a good point. We were aware of this possibility thus we used the same month of ALT measurement to modelled month in our simulations for comparison purposes which minimizes this potential error (as mentioned in our response to Reviewer #1 as well).**

- It appears that most of the developments are detailed in the appendix, but it doesn't appear to include details of the moss parameterisation. Please add something about the changes to thermal/hydraulic parameters for reproducibility (apologies if I missed this).
  **As the moss parameterization is fully detailed in Wu et al. 2016 we had originally referenced that paper. However, for clarity and to be in line with the other parameterizations we test, as suggested, we have now added a brief overview to the appendix.**

- I would expect 'mean absolute error' to be higher for the air than for the soil, because variations in air temperature are typically much larger (both seasonally and on short timescales). Therefore I don't completely agree with the argument on page 27 lines 3-9.
  **One aspect that influences these calculations is temporal averaging. We compute the wMAE based on monthly mean values. The use of monthly means would greatly reduce the impacts of variability. If we did the same comparison at higher time resolutions, we agree that the higher variability of the air temperatures could become a factor.**

**FIGURES**

- I find Figure 3 quite confusing and not entirely helpful. It is not clear what either of the axes are: The residual I guess is obs-model, but it's not specified in the text, and it's not at all clear what the x-axis is showing. Is this the observed ALT? If so I think it would be better, and show the same information, if you just to plot model ALT against observed ALT. I am also not entirely convinced by all the discussions that relate to that figure in terms of 'biases with depth', since there are altogether not many points on the plots and the trends do not appear to be very strong. Perhaps these discussions could be removed or modified to focus on the more significant impacts of the developments. For example, the 20 layer simulation has in general too deep ALT- would be enough information.
  **Since it was peripheral to the main message of the paper, and appeared to confuse roughly 66% percent of readers (Reviewer #1 found it confusing as well), we have removed it.**

- Figure 7 doesn't add much. We can see basically the same thing but more clearly on Figure 4. If it is to be included, I would suggest in the appendix or supplementary.
  **We have moved it to the supplement.**

- Figure 11: I am confused as to why the simulation with moss is warmer than the simulation with no moss. Are these labelled correctly?
  **Thank you for catching this, they were switched. This is now corrected.**

**MINOR COMMENTS**

- "with seasonal wMAE values for the shallow surface layers of the revised model simulation at most 1.2C greater than those calculated for the model driving screen-level air temperature compared to observations at the sites" - This sentence is difficult to
  parse. Perhaps by removing the last part ("compared to observations at the sites"), it would make

more sense.

**We have taken your suggestion and reworded to:** *"with seasonal wMAE values for the shallow surface layers of the revised model simulation at most 3.7 ($^{\circ}$)C, which is 1.2 ($^{\circ}$)C more than the wMAE of the screen-level air temperature used to drive the model as compared to site-level observations (2.5 ($^{\circ}$)C"*

- Page 7 line 14: How is the 1851-1900 part different from just doing two more spinups? Is the CO2 varying? CO2 forcing dataset is not mentioned but should be included - please add.
**Both CO2 and land cover change during that period. We have added explanatory text around this to the revised MS. Both of which have little impact on permafrost physics which is why we neglected it in the first draft.**

- Page 8 line 10: "filtered out"- not sure what that means? Does it mean removing the same values from model as are missing in observations?
**The missing values were removed from further consideration. We have removed this statement as this processing step isn't actually required since missing values aren't considered in calculating monthly means.**

- Please include how you define permafrost presence in a grid cell (again, apologies if I missed this).
**This is on page 7:** *'Active layer thickness in CLASS-CTEM is determined by the temperature and water content of the ground layers. If a layer's temperature is 0($^\circ$C), the frozen water fraction is used to estimate the thickness of freezing within the layer, i.e., if half of the water content in the layer is frozen, the ALT is assumed to be halfway through the layer. Permafrost area in the model domain was calculated by selecting grid cells with active layer thicknesses less than the model total ground column and multiplying by the grid cell area.'*

- Page 9 line 31. 'more grid cells' -> 'additional grid cells' (for clarity)
**Corrected**

- Page 13 line 7-8. I would expect the absence of water and latent heat to make active layer too deep, because latent heat suppresses the seasonal cycle. But since latent heat works equally in both directions (applicable both in freezing and thawing periods),
I am not convinced that its absence would lead to warmer soils. Indeed I am not seeing a major bias in soilgrids vs 20 layers on Figure 6. Perhaps reconsider this sentence.
**This section of the manuscript is comparing the 3 layer model setup to the 20 layer model setup - not yet comparing the 20 layer setup to model configurations with deeper permeable depths. The statement 'The absence of water and therefore of heat consumption by melting ice in these lower ground layers causes the model soil column to be generally too warm.' is then discussing the changes within that context. This particular comparison is complicated by the deepening of the zero flux boundary as is discussed in the same paragraph. Later in the manuscript we compare the 20 layer experiment to the SoilGrids (generally deeper permeable depths) experiment. Here we see generally colder winter and warmer summer ground temperatures in the 20 layers simulation compared to SoilGrids as one would expect due to the latent heat effects of soil water (Figure 6).**

- P 13 line 25: "accurately" -> "accurately simulating"
**Corrected**

- P 15 line 17-18. Soilgrids has extremely deep soils in West Siberia/Urals: Can you find some reference or ask someone who's been there whether this is at all realistic? Having claimed this is a better validated dataset it would be helpful to provide evidence of this.

  **Unfortunately we can only draw upon the evaluation provided in the cited papers. We know of no other data sources available to us.**

- P17 line 15-16. Cold bias due to too much moss is a reasonable assumption, but coupled with the fact that there doesn't seem to be a major warm bias without it (fig 6), and that you would overall expect observations to be biased cold (as written in 3.13), I am not totally convinced here. More consideration of processes and what is missing may be helpful.

  **We can look at this in two ways: (a) the relative change between experiments and (b) bias. The relative change is clear, simulations with moss are colder than those without. The responsible processes are known and described as the thermal offset effect (Goodrich 1982) and as reduced warming by snow in the presence of surface materials with low thermal conductivity (Gruber and Hoelzle 2008). The difference in bias is harder to pin down and can be due to a number of things. Two examples include the inappropriate application (or not) of moss at sites that in reality don't (do) have moss and a bias in the observations available (more borehole non-moss sites than moss sites?).**

- P29 line 21-22 "thus excluding the simulation of taliks". I am confused by this statement. A talik could be simulated in a 1D model. Discontinuous permafrost, on the other hand, couldn't be. Please could you either clarify or rephrase.

  **Yes, this was intended to be 'discontinuous permafrost'. Thanks we have corrected this.**

  Hope you find these comments helpful to improve an already good manuscript.

  **Yes, thank you**

**References Cited:**

**Goodrich, L. E. 1982. "The Influence of Snow Cover on the Ground Thermal Regime." Canadian Geotechnical Journal 19 (4): 421–32. https://doi.org/10.1139/t82-047.**

**Gruber, Stephan, and Martin Hoelzle. 2008. "The Cooling Effect of Coarse Blocks Revisited: A Modeling Study of a Purely Conductive Mechanism." In Proceedings of the 9th International Conference on Permafrost 2008, 557–61. Fairbanks, Alaska, USA.**